# Intercomparison of wind speed, temperature, and humidity data between dropsondes and aircraft in situ measurements

Soodabeh Namdari[1], Sanja Dmitrovic[2], Gao Chen[3], Yonghoon Choi[3,4], Ewan Crosbie[3,4], Joshua P. DiGangi[3], Glenn S. Diskin[3], Richard A. Ferrare[3], Johnathan W. Hair[3], Simon Kirschler[5], John B. Nowak[3], Kenneth L. Thornhill[3,4], Christiane Voigt[5,6], Holger Vömel[7], Xubin Zeng[8], Armin Sorooshian[1,2,8]

[1]Department of Chemical and Environmental Engineering, University of Arizona, Tucson, AZ, 85721, USA
[2]James C. Wyant College of Optical Sciences, University of Arizona, Tucson, AZ 85721, USA
[3]NASA Langley Research Center, Hampton, VA 23681, USA
[4]Analytical Mechanics Associates, Hampton, VA 23666, USA
[5]Institute of Atmospheric Physics, German Aerospace Center, Germany
[6]Institute of Atmospheric Physics, University Mainz, Germany
[7]National Center for Atmospheric Research, Boulder, CO, 30301, USA
[8]Department of Hydrology and Atmospheric Sciences, University of Arizona, Tucson, AZ 85721, USA

*Correspondence to*: Soodabeh Namdari (snamdari@arizona.edu)

**Abstract.** Airborne measurements of wind speed and direction, temperature, and relative humidity are critical due to their importance for atmospheric processes. Field campaigns with multiple coordinated aircraft present challenges when combining data from each platform due to atmospheric heterogeneity. To confront this issue, this work intercompares for the first time in situ measurements from the Turbulent Air Motion Measurement System (TAMMS) of horizontal winds and temperature, and a diode laser hygrometer (relative humidity) deployed on a HU-25 Falcon flying mostly within the marine boundary layer to an independent set of measurements from dropsondes launched from a higher-flying King Air. Leveraging data from 162 joint flights over the northwest Atlantic from these two spatially coordinated aircraft during the NASA ACTIVATE campaign in winter and summer seasons between 2020-2022, a total of 555 pairs of Falcon-dropsonde data points are identified within 30 km horizontal separation, minimal vertical separation (usually < 1 m), and within 15 minutes. This analysis is based on the following range of conditions experienced: altitude = ~0.1-5 km; temperature = -19 – 27 °C; relative humidity = 1 – 100%; wind speed = 0.2 – 42 m s$^{-1}$. Based on scatterplots, correlation coefficients, and mean (in situ – dropsonde) error (ME), intercomparisons reveal good agreement for wind speed (r = 0.95, ME = 0.21 ± 1.68 m s$^{-1}$), u/v wind components (r ~ 0.96-0.97, ME ~ 0.03 – 0.16 (± 1.62 – 1.67) m s$^{-1}$), wind direction (r = 0.94, ME = 0.00 ± 0.22 based on cosine of direction angles), temperature (r = 0.99, ME = 0.00 ± 0.71 °C), and relative humidity (r = 0.91, ME = -3.86 ± 10.74%). Sensitivity analysis shows that binning data into categories of horizontal separation distance, clear versus cloud, winter versus summer, altitude range, and terciles of the values for examined variables did not yield major changes except for relative humidity where there was more deviation especially above 70%. The effect of statistics was examined by relaxing the vertical separation distance criteria to expand the number of pairs to over 360,000, without much difference in intercomparison metrics. The effect of averaging more points for each instrument in the final 555 pairs was also shown to lead to minimal change in agreement.

Overall, these results provide confidence in both the performance of the measurement techniques compared and combining dropsonde data with in situ data from a separate coordinated aircraft for ACTIVATE, which has relevance to other campaigns with multiple coordinated aircraft conducting similar types of measurements.

## 1 Introduction

Wind speed and direction, temperature, and humidity are among the most important features of the atmosphere driving processes such as heat, momentum, and chemical fluxes, gas and particle spatial distributions, and cloud evolution (Lewis & Schwartz, 2004; Nuijens & Stevens, 2012; Neukermans et al., 2018). These types of data are commonly obtained using surface measurements and satellite remote sensing, with airborne measurements being a critical link between the two (e.g., Lenschow, 1986; Thornhill et al., 2003). Airborne measurements of these state variables, especially wind speed and direction, are challenging due to changes in airflow around aircraft, which warrants calibrations only in flight rather than on the ground where the same conditions cannot easily be simulated with air flowing past an airplane. There have been very limited reports of cross-comparisons between air motion system instruments on aircraft flying next to each other (e.g., Lenschow et al., 1991; MacPherson et al., 1992; Thornhill et al., 2003). Those studies relied on similar types of instruments on aircraft flying in close formation and are important to gain confidence in reported data. It is critical to assess performance for techniques in airborne field work that are the focus of this work including dropsondes, the Turbulent Air Motion Measurement System (TAMMS) and the diode laser hygrometer (DLH). Dropsondes can be specifically compared to TAMMS for wind speed/direction (including the u and v components of wind) and temperature (T) and to DLH for relative humidity (RH). Hereafter, references to wind speed refer to $\sqrt{u^2 + v^2}$ whereas references to u and v components will be explicitly mentioned. Furthermore, a growing number of field campaigns like ACTIVATE, CAMP$^2$Ex, and ARCSIX involve multiple coordinated aircraft with one platform launching dropsondes that is spatiotemporally removed from other aircraft. The dropsonde data and other datasets from the same aircraft are used together with the other aircraft for various scientific applications, but the question remains as to how valid this exercise is due to heterogeneity in atmospheric conditions. Intercomparisons between atmospheric state variables between the different platforms can provide a view of how well this strategy can work. If such variable (e.g., winds, temperature, humidity) values from independent measurements agree well between the two aircraft in a complex environment like the northwest Atlantic (subject of this study), data from both aircraft can be used together in a meaningful way for certain scientific applications.

To provide context, Table 1 provides a summary of intercomparison studies, highlighting the methods used to validate wind speed/direction, temperature, and humidity measurements. While the primary focus of this study is the unique comparison of dropsonde data with independent in situ measurements of similar variables from another aircraft, intercomparisons based on other techniques are included in the table to provide important historical context, demonstrating how different datasets have been compared and validated across platforms. These studies fall into the following categories of how intercomparisons were

conducted: (i) airborne lidar versus dropsondes launched from same aircraft; (ii) airborne radiometer versus dropsondes launched from same aircraft; (iii) airborne lidar versus surface measurements; (iv) multiple aircraft in close proximity with separate air motion systems; and (v) unmanned aerial vehicle measurements versus surface measurements. For instance, Baidar et al. (2018) compared wind measurements obtained from the Green Optical Autocovariance Wind Lidar (GrOAWL) with co-located dropsonde measurements during their validation process aboard the NASA WB-57 research aircraft. The campaign

involved several flights over the Gulf of Mexico. The GrOAWL successfully measured winds with high accuracy compared to dropsondes, demonstrating excellent agreement ($R^2 > 0.9$). Bucci et al. (2018) compared wind speed observations from an airborne doppler wind lidar (ADWL) during tropical cyclones to dropsondes and reported correlation coefficients of 0.96 and 0.92 for wind speed and wind direction, respectively. The average separation distance between the paired datasets was ~2 km (maximum of 5 km and 6 minutes of separation). Thornhill et al. (2003) compared air motion systems onboard two aircraft

(NASA P3-B and NCAR EC-130Q) over the Sea of Japan during spring 2001 as part of the TRACE-P and ACE-ASIA missions. Findings indicated a high degree of agreement between the two aircraft systems, particularly in measuring lower-frequency components of winds and potential temperature; the regression slopes for the mean wind components and potential temperature were nearly unity between the two aircraft. However, a limitation in that study was the use of data in only one research aircraft flight on 2 April 2001 due to stringent requirements to ensure aircraft were within 200 m and 10 m in the

horizontal and vertical directions, respectively, as done in other works too (Nicholls et al., 1983). Table 1 reveals that these intercomparisons were usually based on very few flights for those studies involving airborne platforms, which is an aspect this current study aims to improve upon along with providing the first report of how an air motion system compares to dropsondes launched from a different aircraft. Table 1 is not exhaustive of all intercomparison studies for atmospheric state parameters and that other studies have examined factors such as temperature (e.g., Barbieri et al., 2019) and water vapor mixing ratio

(Vömel et al., 2007; Weinstock et al., 2009; Kaufmann et al., 2018), where small differences (i.e., < 1 ppm) can have major implications in the stratosphere for the radiation budget (Solomon et al., 2010).

The goal of this study is to use a unique and comprehensive dataset from the NASA ACTIVATE field campaign to compare measurements of temperature, relative humidity, and wind speed/direction between various instruments on an aircraft flying mostly in the boundary layer and dropsondes launched from a separate higher-flying aircraft spatially coordinated at ~9 km.

If the results show good agreement, this provides confidence in both the performance of the independent measurement techniques and confidence in integrating dropsonde data with the Falcon for specific scientific applications especially if there are no Falcon data for atmospheric state variables on a given flight such as when icing impacts data quality. This paper is structured as follows: Sect. 2 provides a summary of ACTIVATE, relevant instrumentation, and the co-location method to compare airborne and dropsonde data; Sect. 3 reports results of the intercomparisons of wind speed/direction, temperature,

and relative humidity; and Sect. 4 states conclusions.

**Table 1: Summary table of past intercomparison studies for atmospheric state variables using observational platforms. Quantitative findings are reported in the 4th column when they were available in the given text of a particular reference.**

| Type of Intercomparison | Reference | Data Compared, Measurement Notes, and Co-location Details (if provided) | Findings | Study Area: Campaign | Number of Comparisons |
|---|---|---|---|---|---|
| Aircraft Lidar VS Dropsonde (same aircraft for both) | Weissmann et al., 2005 | • Wind Speed Profile<br>• Dropsonde measurements averaged into 100 m vertical segments to match the lidar's 100 m vertical resolution<br>• Horizontal separation ranged from 0 to 23 km (mean = 6.5 km)<br>• Time difference up to 11 minutes | Standard deviation of the difference between dropsonde and lidar wind speeds ranged from 0.6 to 1.8 m/s. The mean difference (bias) between the lidar and dropsonde wind speeds was -0.31 m s$^{-1}$. | • Atlantic Ocean<br>• Atlantic "The Observing System Research and Predictability Experiment" (THORPEX) Regional Campaign (A-TReC) | 8 flights, 33 dropsondes |
| | Kavaya et al., 2014 | • Wind Speed Profile<br>• ~10 km measurement separation<br>• Time separation ~ 13 min for one intercomparison | Notes of "very good" and "excellent" agreement in wind measurements. | • Marine areas by U.S. and Mexico<br>• Genesis and Rapid Intensification Processes (GRIP) campaign | 1 flight, 1 dropsonde |
| | Baidar et al., 2018 | • Wind Speed Profile<br>• Dropsonde-measured winds were projected onto the lidar line of sight, interpolated to the center of the lidar measurement grid, and compared to the closest lidar measurement in that location | High correlation ($R^2 > 0.9$) between wind speeds | • Gulf of Mexico<br>• Atmospheric Transport,<br>• Hurricanes, and Extratropical Numerical Weather Prediction with the Optical Autocovariance Wind Lidar (ATHENA-OAWL) mission | 8 flights, 44 dropsondes |
| | Bucci et al., 2018 | • Wind Speed/Direction Profile<br>• Horizontal separation < 5 km (mean = 2.1 km)<br>• Temporal separation < 6 min | High correlation between lidar and dropsonde wind speeds (r=0.96) and direction (r=0.92). | • Marine areas by U.S. and Mexico | 5 flights, 49 dropsondes |
| | Dmitrovic et al., 2024 | • Near-surface Wind Speed<br>• Horizontal separation <30 km<br>• Temporal separation < 15 min | Lidar accuracy = 0.15 ± 1.80 m s$^{-1}$. High correlation between wind speed (r: 0.89/0.88 for two different wave-slope vs. wind speed parameterizations). | • Northwest Atlantic<br>• ACTIVATE (same as this study) | 168 flights (90 in winter, 78 in summer), 577 dropsondes |
| | Bedka et al., 2021 | • Wind Speed/Direction Profile, Water Vapor Mixing Ratios (including from DLH), Aerosol Backscatter Profiles<br>• Dropsonde wind data within ±33 m of each lidar altitude bin<br>• Lidar wind profile obtained within 2.5 minutes of the sonde release<br>• Water vapor mixing ratios and aerosol backscatter profiles data were averaged over 10 to 60 seconds (horizontally approximately 6–12 km) depending on the flight speed | Strong agreement (lidar vs dropsonde) between wind speed/direction: Precision (i.e., RMSD) = 1.22 m s$^{-1}$/7.6 °; accuracy (i.e., bias) = 0.12 m s$^{-1}$/1.02 °.<br>Good agreement for water vapor mixing ratio (average percent difference of 5%). | • Eastern Pacific Ocean<br>• NASA Aeolus Calibration/Validation (Cal/Val) Test Flight campaign | 5 flights, 61 dropsondes |
| Aircraft Radiometer VS Dropsonde (same aircraft for both) | Cecil & Biswas, 2017 | • Wind Speed<br>• Horizontal separation < 500 m | Generally good agreement reported with root mean square differences around 4.1 – 6.3 m s$^{-1}$ | • Atlantic Ocean, including Hurricanes Joaquin, Patricia, and the remnants of Tropical Storm Erika<br>• Tropical Cyclone Intensity (TCI) experiment | Multiple flights (did not specify the number of flights), 636 dropsondes |
| Aircraft Doppler Lidar VS Surface Measurements | Reitebuch et al., 2001 | • Wind Speed/Direction Profile<br>• Ground site includes wind profiler radar and radiosondes<br>• Flight time over surface station | Generally good agreement reported | • Lindenberg, Germany | 1 flight |

| | | | | | |
|---|---|---|---|---|---|
| | De Wekker et al., 2012 | • Wind Speed/Direction Profile<br>• Ground site includes wind profiler radar and radiosondes<br>• Flight time over surface stations | Generally good agreement reported | • Salinas Valley, California | 1 flight |
| Multiple Aircraft in Close Proximity with Separate Air Motion Systems | Lemone & Pennell, 1980 | • 3-D Wind, Temperature, Moisture, and Others<br>• Aircraft: C-130, DC-6, L-188<br>• Horizontal separation <50 m<br>• Vertical separation < 10 m | Generally good agreement reported | • Tropical Atlantic<br>• Global Atmospheric Research Programme (GARP) Atlantic Tropical Experiment (GATE) | 7 flights |
| | Nicholls et al., 1983 | • Means, Variances, Spectra/Co-Spectra of 3-D Winds<br>• Aircraft: C-130, Electra, Falcon<br>• Horizontal separation <200 m<br>• Vertical separation <10 m | High degree of agreement reported | • North Atlantic<br>• Joint Air-Sea Interaction (JASIN) Experiment | 5 flights |
| | Lenschow et al., 1991 | • Means, Variances, Spectra/Co-Spectra of 3-D Winds, Water Vapor Mixing Ratio (WVMR), Temperature<br>• Aircraft: Electra, Sabreliner, King Air<br>• Horizontal separation <30 m | Generally good agreement reported | • N/A | 2 flights |
| | MacPherson et al., 1992 | • Wind Speed/Direction, Temperature, Humidity, and Others<br>• Aircraft: 2 King Airs, 1 Twin Otter | Generally good agreement, with agreement in wind component measurements typically within 1 m s$^{-1}$.<br>Temperature agreement within 0.3°C; RH variability <2 g kg$^{-1}$. | • Kansas<br>• First International Satellite Land Surface Climatology Project (ISLSCP) Field Experiment (FIFE) | 7 flights |
| | Thornhill et al., 2003 | • Means, Variances, Spectra/Co-Spectra of 3-D Winds, Temperature<br>• Aircraft: P3-B, EC-130Q<br>• Horizontal separation <200 m<br>• Vertical separation <10 m | Generally excellent agreement with scatterplot slopes and y-intercepts of 1 and 0, respectively. | • Sea of Japan<br>• TRACE-P and ACE-ASIA | 1 flight |
| Unmanned Aerial Vehicle Measurements VS Surface Measurements | Martin et al., 2011 | • Wind Speed/Direction, Temperature, and Humidity<br>• UAV compared to in situ and ground-based remote sensing systems<br>• Spatial separation ~ 5 km | Generally good agreement. Temperature differences < 1 K; RH differences < 20%. | • Lindenberg, Germany | 2 flights on 1 day |
| | Witte et al., 2017 | • Wind Speed/Direction, Temperature, Relative Humidity<br>• Unmanned Aerial Vehicle (UAV) - BLUECAT5 vs. Ground-based anemometer at 7.62 m height, and Oklahoma Mesonet Network Tower<br>• UAVs flown at 40-120 m AGL | Generally good agreement reported | • Oklahoma, USA. | 3 flights on same day |
| | Wetz & Wildmann, 2022 | • Wind Speed/Direction<br>• Fleet of unmanned aircraft systems (UAS) vs ground-based sonic anemometers<br>• Measurements aligned with sonic anemometers at 50 m and 90 m altitudes<br>• Minimum horizontal separation distance of 20 m | High accuracy in wind measurement (RMSE = 0.25 m s$^{-1}$) for horizontal wind speed and wind direction (RMSE < 5°) | • Falkenberg, Germany | 119 |
| | Scicluna et al., 2023 | • Wind Speed/Direction<br>• Remotely Piloted Aircraft System (RPAS) vs ground-based lidar<br>• Drone operation over surface site<br>• RPAS operated ~75 m southeast of the lidar at altitudes of 40, 60, 80, and 100 m above ground, with minimum 5 min uninterrupted hover to ensure alignment within the lidar's measurement window | High wind speed correlation ($R^2$ = 0.90; Slope = 1.01, y-intercept = 1.00) and wind direction correlation ($R^2$ = 0.95; Slope = 1.00, y-intercept = -6.16) | • Ċirkewwa, l/o Mellieħa, Malta | 40 flights |

## 2 Methods

### 2.1 ACTIVATE Campaign Description

The NASA Aerosol Cloud meTeorology Interactions oVer the western ATlantic Experiment (ACTIVATE) aimed to provide
an important dataset to advance knowledge of aerosol-cloud-meteorology interactions including both improved
parameterization of these processes in climate models and technological advances for instrumentation and retrievals of related
geophysical variables (Li et al., 2022; Schlosser et al., 2022; Sorooshian et al., 2023). The project involved airborne flights
over the western North Atlantic in different seasons of each year between 2020-2022 to generate a dataset covering different
weather regimes (and thus cloud types) and aerosol types depending on the season. The flights are categorized into winter and
summer seasons, but note that these categories are loose in that winter covers November-April and summer spans May-
September (Sorooshian et al., 2023). Most flights were based out of NASA Langley Research Center (LaRC) in Hampton,
Virginia. In June 2022, 17 flights were conducted based out of Bermuda for an intensive period of operations to extend the
spatial range of ACTIVATE flights farther offshore. ACTIVATE involved a disciplined approach of conducting spatially
coordinated flights with NASA Langley's HU-25 Falcon and King Air aircraft. The flight strategy coordinated a King Air
flying at ~9 km and Falcon flying stair steps at various levels below 3 km, including below, in and above boundary layer
clouds. The coordinated approach allowed for near-simultaneous sampling of trace gases, aerosols, cloud microphysics, and
atmospheric state parameters from both in-situ and remote sensing techniques between near the ocean surface and ~9 km. Over
90% of the flights were able to utilize this coordinated approach and serve as the basis for the analysis in this paper. The
primary instruments used in this study are dropsondes from the National Center for Atmospheric Research (NCAR) Airborne
Vertical Atmospheric Profiling System (NCAR AVAPS) instrument that was flown on the King Air and both the TAMMS
and DLH instruments on the Falcon.

### 2.2 King Air: Dropsondes

The NCAR AVAPS (NSF-NCAR, 1993) was used on the King Air to control dropsonde operations. The project utilized the
NCAR designed and built  NRD41 sondes in lieu of the older RD41 model due to improved launch detection and electronics.
During the three years of operations, a total of 801 sondes were launched, with 788 (98%) providing a complete profile from
the aircraft altitude all the way to the surface. They are the ones that will be used in this study.  Data were processed using
NCAR's Atmospheric Sounding Processing Environment (ASPEN; https://www.eol.ucar.edu/content/aspen; Vömel et al.,
2023). ASPEN ensures the integrity and accuracy of the data by applying dynamic corrections and smoothing algorithms,
enhancing the reliability of the atmospheric profiles. The dropsonde data underwent a series of corrections and quality control
steps to ensure accuracy and reliability, following the methods detailed in Vömel et al. (2023) and Aberson et al. (2023). These
included dynamic corrections to account for the lag in temperature and humidity sensors, with an equilibration time of 10

seconds applied to eliminate artifacts after release from the aircraft. Wind data were corrected for sonde inertia to address motion-induced errors during descent. A B-spline smoothing algorithm was employed for pressure, temperature, humidity, and wind components, with a 5-second smoothing window to maintain profile continuity while preserving critical gradients.

Post-smoothing, wind speed and direction were recalculated to ensure consistency. Additionally, outliers and suspect data points were removed, and surface pressure values were extrapolated using fall rates, followed by recalculating geopotential height and vertical wind velocity. Any telemetry issues, such as synchronization errors between the onboard computer and GPS, were addressed through post-processing and reconstruction of raw data. These steps collectively ensured high-quality datasets suitable for robust analysis (Aberson et al., 2023; Vömel et al., 2023).

The final science quality data archive includes vertical distributions of temperature, pressure, humidity, and winds (u, v, w components) at the sub-hertz acquisition rate at which they were required. The reported uncertainty in the zonal wind components is 0.5 m s$^{-1}$ (Sorooshian et al., 2023). Uncertainties in the temperature and relative humidity measurements are 0.2 °C and 3%, respectively. The vertical resolution of the dropsonde data is approximately ~6 m. Relative humidity intercomparisons are conducted in this work between dropsondes and the Falcon only when dropsonde relative humidity was

below 100%. Customarily, relative humidity measurements above 100% are set equal to 100% in quality controlled data (Vömel et al., 2023) to eliminate potentially erroneous measurements due to supersaturation and the potential presence of liquid water that could introduce biases into the analysis. Typically, raw measurements may exceed the 100% level by a few percent due to measurement uncertainty at saturation and due to environmental factors, mostly the presence of liquid water in clouds. However, this study excludes all dropsonde relative humidity values ≥ 100%, which represents less than 5% of the

data with negligible impact on final results. Intercomparisons among variables other than relative humidity are still conducted if dropsonde relative humidity ≥ 100%. Details on the uncertainty analysis for dropsondes, including sensor response times and calibration procedures, are provided in the Supplementary Material file (Sect. S1). This includes estimates for pressure, temperature, relative humidity, and wind speed uncertainties as supported by literature (Bärfuss et al., 2018; Bärfuss et al., 2023; Vömel et al., 2023).


### 2.3 Falcon Measurements

Fast response, high frequency measurements of the 3-dimensional wind field (horizontal (u and v) and vertical (w)), static air temperature, and pressure as well as position and altitude are provided from the NASA TAMMS instrument on the Falcon. Wind component measurements are derived from an inertial navigation system (Applanix 610) to provide the speed of the

aircraft with respect to the Earth and a combination of pressure and temperature measurements used to derive the speed of the air with respect to the aircraft. The difference in those two vectors provides the ambient 3-dimensional wind field. The static and dynamic pressures are sampled with Honeywell PPT2 pressure sensors and the Total Air Temperature (TAT) is measured with a Rosemount 102 non-deiced TAT sensor. The differential pressures needed for the angles of sideslip and attack are computed from five flush-mounted ports on the radome of the Falcon arranged in a cruciform pattern. Utilizing the radome

minimizes any airflow biases around the aircraft. Extensive calibrations done over the course of the three flight years accounted for well determined calibration coefficients for the angles of attack and sideslip and a good characterization of the heading offset and pressure defect necessary to compute high precision ambient winds. Focus in this study is on the TAMMS horizontal wind component measurements. Ambient wind component data derived from aircraft are sensitive to deviations away from straight and level flight conditions and artificial artifacts can be introduced. As such, data usage is restricted to times when the

Falcon is flying straight and level (pitch and roll angles less than 5 degrees (absolute)). Similar to dropsondes, the reported uncertainty in the zonal wind component is 0.5 m s$^{-1}$ (Sorooshian et al., 2023). A full explanation of the calibration maneuvers for wind component measurements is provided in the Supplementary Material (Sect. S1). Temperature measurements associated with the TAMMS have an uncertainty of 0.5 °C. More details about the TAMMS can be found elsewhere (Thornhill et al., 2003; Sorooshian et al., 2023). It has been used on the NASA P-3 aircraft for over 25 years, and this was the first time

it was used on the Falcon.

     Relative humidity is derived from water vapor measurements using a DLH, which has an uncertainty of 5% or 0.1 ppmv (Sorooshian et al., 2023). Relative humidity calculations also depend on temperature and pressure measurements, introducing additional uncertainties that propagate into the final values (Garcia Skabar, 2015). The saturation vapor pressure, determined by temperature, is particularly sensitive, with small temperature uncertainties leading to noticeable relative humidity variations.

The impact of pressure measurement uncertainties is relatively small, as the uncertainties represent a very small fraction of pressure itself. As a result, the combined uncertainty of relative humidity is higher than that of the water vapor mixing ratio (WVMR), depending on conditions. References to relative humidity in this work refer to the relative humidity with respect to liquid water. The DLH is an open-path and near-infrared absorption spectrometer (Diskin et al., 2002) with its open path completely on the Falcon's exterior between a retroreflector on the starboard wing and a cabin window. The time resolution

of the TAMMS and DLH is 0.2 s (or ~25 m in horizontal distance) or better, but data are averaged to 1 s resolution in this work.

## 2.4 Data Processing and Statistical Methods

     A key component of the analysis was finding the best spatial and temporal match between data collected from dropsondes and

the Falcon. During joint flights, the two aircraft were co-located within 5 min and 6 km approximately 73% of the time (Sorooshian et al., 2023), providing a relatively large sample set for intercomparison. However, due to the inhomogeneity of the northwest Atlantic and the fact that dropsondes are carried along the mean wind direction and do not descend straight down from the time they are launched to the time they reach the altitude of the Falcon, rugged constraints must be used in order to get closer to a fair comparison. Figure 1 summarizes the criteria applied to identify the best match between the Falcon

measurements for a given dropsonde launched from the King Air altitude down to the ocean surface, which ideally should be done for other campaigns involving multiple coordinated aircraft involving dropsonde launches. First, dropsondes and the Falcon had to be co-located within 30 km, vertically within 25 m, and within 15 min to strike a balance between having a good

sample size for analysis and reduce the chances of the two measurements reflecting very different dynamical regimes. These criteria (except the vertical requirement) were also used in the recent wind intercomparison analysis by Dmitrovic et al. (2024) using ACTIVATE data (Table 1). For context, the Falcon's air speed was typically 120 m s$^{-1}$, and 30 km represents 250 s of flight.

The calculation of distances between points was conducted using the haversine formula, which is needed for determining the great-circle distances on a spherical surface based on longitude and latitude (Hoover et al., 2013; Schlosser et al., 2024). As noted in the previous section, this analysis only uses wind component data collected during straight and level flight legs. Intercomparisons are conducted only in cases when the two instruments were either both in cloud or both out of cloud (i.e., avoid one in cloud and the other in clear air). Criteria for identifying whether the TAMMS was in cloud relied on the Fast Cloud Droplet Probe (FCDP; Spec Inc.), which measured aerosol and cloud droplet size distributions between 3 and 50 µm (Kirschler et al., 2022; Kirschler et al., 2023). Cloud sampling was based on both liquid water content (LWC) exceeding 0.01 g m$^{-3}$ and cloud drop number concentration ($N_d$) exceeding 10 cm$^{-3}$. Conversely, clear-sky conditions were defined by LWC values less than 0.001 g m$^{-3}$ and $N_d$ being below 5 cm$^{-3}$. For dropsonde measurements, criterion for being in and out of cloud included when relative humidity was above 97% and below 94%, respectively.

After those criteria were met, candidate pairs emerge between the dropsonde and the Falcon whereby the final pair was based on finding the minimum altitude difference between the two. This was done with the notion that a better intercomparison is possible for cases with independent measurements being at a closer altitude versus a closer spatial distance. For instance, temperature is much more sensitive to vertical changes rather than horizontal changes in the atmosphere. A total of 555 pairs were identified after all these criteria were applied to all dropsonde data points and all Falcon data points (locations in Figure 2). For some geophysical variables compared in this work, there were times when no data were collected by a particular instrument and thus there were not a full set of 555 pairs to compare. Pairs of data points are additionally examined using the method in Figure 1 excluding the final step of finding just one pair of data points per dropsonde launch with the minimum altitude difference to allow for a much larger population of data pairs (>360,000 pairs). Additional analysis in this paper examines how intercomparisons changed when using more points collected before and after those respective single points.

For context, Figure 3a-b shows histograms for the distribution in the vertical and horizontal distances between the final data points comprising the 555 pairs. While this study's criteria allows up to 25 m, the majority of the pairs (~90%) exhibit vertical differences of less than one meter because of the final step to use a pair of points with the smallest vertical gap while adhering to the other criteria. This observation highlights the stringent nature of the selection process, where minimizing the altitude difference is an important factor due to the sensitivity of atmospheric state variables to altitude variations. The median horizontal separation distance between pairs was 18.8 km. The median geopotential altitudes of the final pairs were (Falcon/dropsonde) 966/965 m with minimum and maximum values of 112/112 m and 4948/4937 m, respectively (Figure 3c-d).

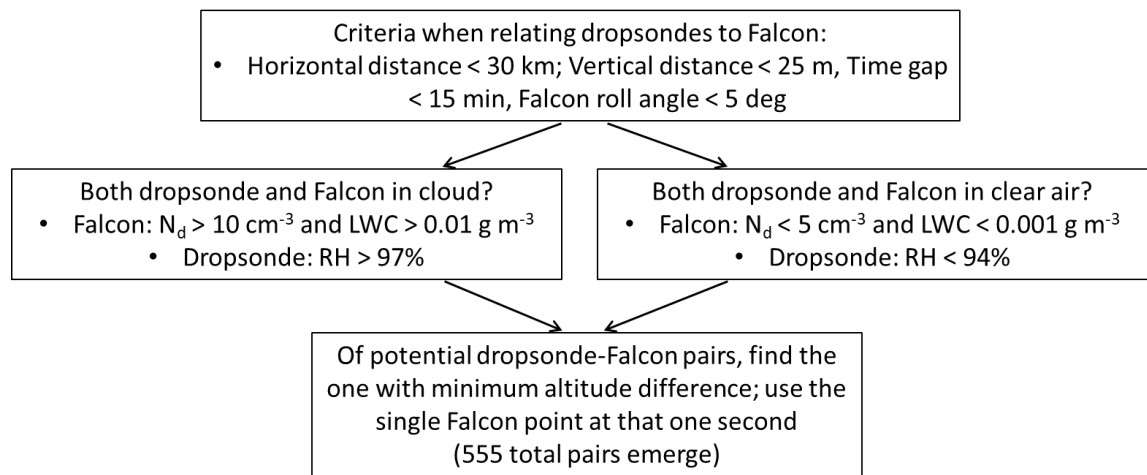

**Figure 1: Methodology of finding the closest points between the Falcon and dropsondes launched by the King Air during ACTIVATE 2020-2022 flights.**


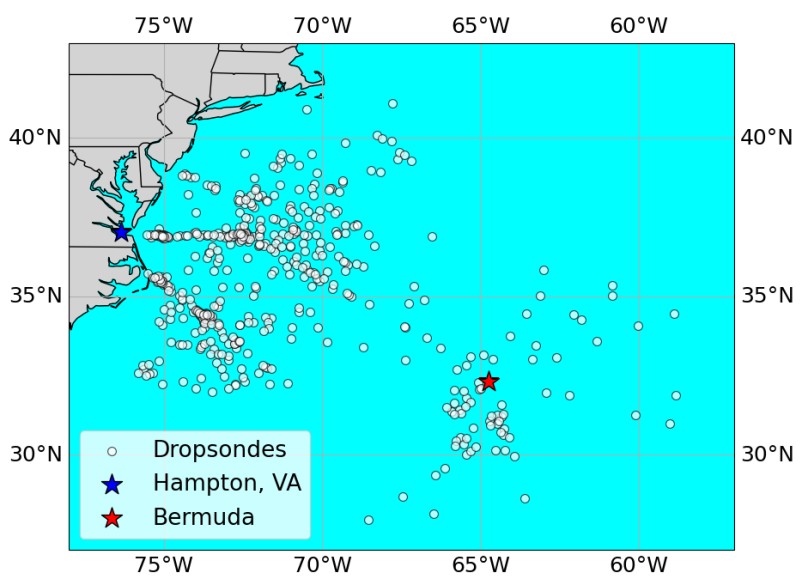

**Figure 2: Spatial map of dropsondes released from the King Air that are utilized for intercomparison with the Falcon measurements. A total of 555 dropsondes are shown that are used after the data screening procedure in Figure 1.**

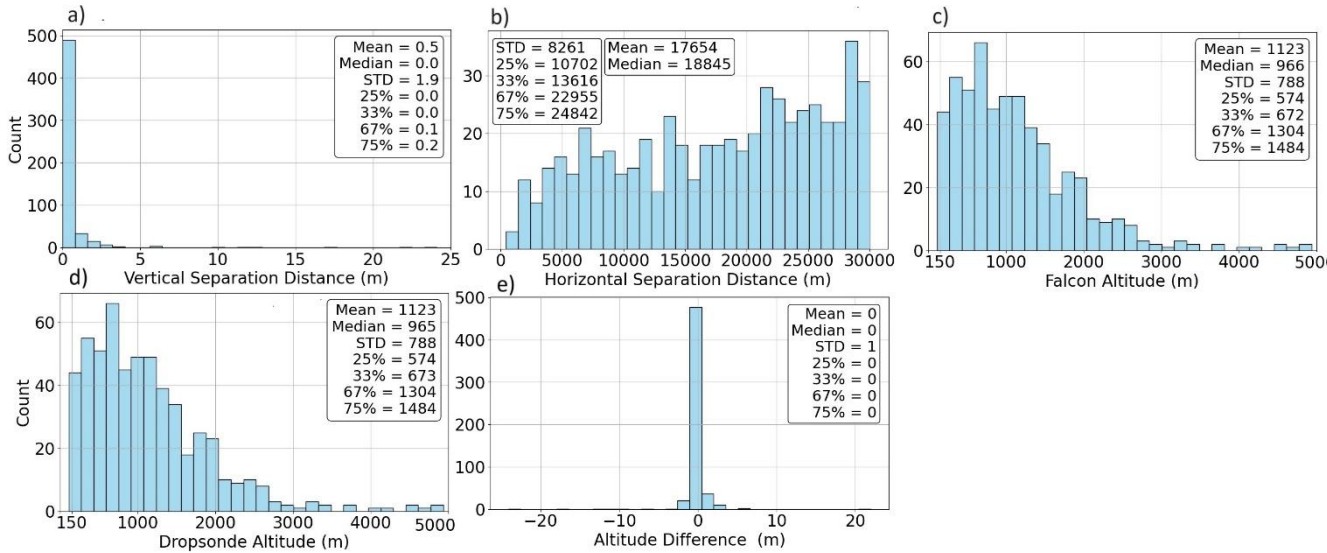

**Figure 3: Histogram showing the (a) vertical separation distance and (b) horizontal separation distance between the final 555 Falcon-dropsonde pairs. Also shown are histograms of the final altitude for each pair of points for (c) the Falcon and (d) dropsondes, and (e) altitude difference between datasets: Falcon - dropsonde.**

In this study, geopotential altitude is used for intercomparison between the Falcon and the dropsonde measurements. The analysis comparing the dropsonde and the Falcon measurements relies on intercomparisons in the form of scatterplots and statistical metrics that closely follow the related work of Dmitrovic et al. (2024), which intercompared ocean surface wind speeds using dropsondes and the High Spectral Resolution Lidar – generation 2 (HSRL-2) during ACTIVATE, with the HSRL-2 having been on the higher-flying King Air too. Single points are compared between the dropsondes and the Falcon that

comprise the final 555 pairs, but those 555 pairs are also intercompared using more data points before and after the single points identified using the process shown in Figure 1. More specifically, a subset of results relies on taking the mean of the single dropsonde point and the preceding and subsequent points (for a total of 3 points and thus ~30 m of vertical distance). Analogously, various numbers of points (2, 5, 10) are used before and after the identified single Falcon point amounting to a total of 5, 11, and 21 points (i.e., ~0.6 km, ~1.3 km, ~2.5 km horizontal distance, respectively) used in averaging. Presented

are correlation coefficients (r), linear best-fit regressions, and geometric mean bisector regressions (hereafter referred to as bisector regressions). The bisector technique differs from the linear regression in that for the former there is no clear distinction between dependent and independent variables and so both the x and y variables are subject to error (e.g., Ricker, 1973); in contrast, for linear regressions only the y variable is the dependent variable, which arbitrarily is chosen to be the Falcon data in this study while the dropsonde measurements represent the independent variable. In this analysis, standard error of the slope

is used to assess the precision of the regression model. Standard error helps assess how close the regression line fits the data and gives insight into the variability or confidence in the slope estimate. A lower standard error signifies a more reliable slope

estimate. Histograms are also presented to provide a visual sense of the distribution and frequency of values in various measurements and calculations.

For intercomparisons, mean error (ME) is computed to quantify discrepancies between the two measurement methods. Mean error is the average difference between the measurements taken by the Falcon instruments (TAMMS and DLH) and dropsondes:

$$\text{Mean Error} = \frac{1}{N}\sum_{i=1}^{N}(\text{Falcon}_i - \text{Dropsonde}_i) \qquad \text{(Eq. 1)}$$


A low ME suggests better agreement while a higher ME shows higher systematic bias between the instruments. A positive mean error indicates that the TAMMS/DLH measure the higher values relative to the dropsondes. Standard deviation is also reported alongside the mean error values to quantify how spread out the errors are around the mean error. A low standard deviation means that the errors are consistently close to the mean error.

It is cautioned that due to the nature of comparing dropsondes launched from an aircraft distinct from another one doing independent measurements, intercomparisons are conducted with the possibility of natural horizontal variability in the variables compared. For context, Table 2 shows the typical standard deviation for temperature, relative humidity, and wind speed (along with u/v components) for level leg types flown by the Falcon during ACTIVATE. To put the standard deviations in perspective, the median value of each variable is calculated for a given leg and then the mean of those medians is obtained

for that same leg type for a given season. The reported standard deviations in Table 2 also represent the mean value of all the standard deviations for individual legs of a given leg type and season. Typical legs were ~3.3 min long and covered almost 25 km horizontal distance (Dadashazar et al., 2022), which exceeds the aforementioned median separation distance (18.8 km) between the Falcon and dropsondes for the final 555 data pairs in this study. The analysis in Table 2 was conducted for cloud-free conditions using the criteria provided already. As shown, standard deviations for given leg types and seasons were usually

less than 0.3°C for temperature, less than 5% for relative humidity, and less than 1 m s$^{-1}$ for the wind variables, which is comparable to the uncertainties in these measurements. Based on Table 2, the intercomparisons presented subsequently are found to be meaningful.

Furthermore, showcasing data across different seasons over three years naturally presents differences in atmospheric stability and boundary layer structure, which is relevant to consider in this study focused on intercomparisons of temperature, humidity,

and wind speed and direction. For context, recent work relying on 506 dropsondes launched during ACTIVATE (Xu et al., 2024) revealed that the median cloud fraction for ACTIVATE flights was 0.22 with the median planetary boundary layer height for cloud fractions below and above 0.22 being 659 m and 1,169 m, respectively. Further, ~28% of those dropsondes indicated the presence of decoupled boundary layers. In light of this type of variability, steps are still taken to enhance meaningful intercomparisons between dropsondes and in situ data such as accounting for separation in time and

horizontal/vertical distances along with comparing data between similar seasons and when clouds are present or not.

**Table 2. Summary statistics (median and standard deviation) for temperature (T), relative humidity (RH), and winds (including u and v components) based on Falcon measurements in specific legs flown (ABL = above boundary layer top; BBL= below boundary layer top; ACT = above cloud top; BCT = below cloud top; ACB = above cloud base; BCB = below cloud base; MinAlt = minimum altitude flown and typically ~150 m above sea level ). These legs were typically between 3-4 min long and the median and standard deviations of each leg were then compiled with the final values shown below being the mean value of all those median and standard deviation values of all legs for a particular leg type and season (shown in format of "summer value/winter value"). The farthest right column represents the number of legs used for the calculations.**

| Leg Type | WS Median ($\mathrm{m\,s^{-1}}$) | WS STD ($\mathrm{m\,s^{-1}}$) | u Median ($\mathrm{m\,s^{-1}}$) | u STD ($\mathrm{m\,s^{-1}}$) | v Median ($\mathrm{m\,s^{-1}}$) | v STD ($\mathrm{m\,s^{-1}}$) | T Median (°C) | T STD (°C) | RH Median (%) | RH STD (%) | No. Legs |
|---|---|---|---|---|---|---|---|---|---|---|---|
| ABL | (7.20/10.43) | (0.42/0.58) | (3.75/7.54) | (0.42/0.58) | (0.25/-0.27) | (0.44/0.62) | (17.66/5.22) | (0.16/0.28) | (61.36/30.77) | (2.57/3.71) | (218/163) |
| BBL | (7.04/8.83) | (0.48/0.76) | (2.82/4.95) | (0.48/0.77) | (0.37/1.26) | (0.52/0.75) | (19.01/6.45) | (0.19/0.23) | (67.28/58.54) | (3.98/5.80) | (140/105) |
| ACT | (7.14/12.79) | (0.53/0.75) | (1.73/9.61) | (0.53/0.80) | (-0.92/-1.22) | (0.56/0.73) | (13.63/-0.35) | (0.24/0.38) | (56.35/28.54) | (4.13/4.18) | (194/295) |
| BCT | (7.03/10.22) | (0.81/1.03) | (0.58/5.51) | (0.78/1.02) | (-0.68/-1.30) | (0.82/1.03) | (14.21/-2.21) | (0.29/0.39) | (80.88/78.07) | (5.88/8.93) | (169/266) |
| ACB | (7.50/9.67) | (0.63/1.04) | (0.19/3.86) | (0.62/1.06) | (0.21/-2.55) | (0.67/1.07) | (18.49/3.06) | (0.14/0.20) | (86.03/81.39) | (3.00/5.05) | (409/607) |
| BCB | (7.29/9.59) | (0.62/1.01) | (0.98/4.45) | (0.63/1.04) | (0.19/-1.95) | (0.68/1.02) | (17.10/0.94) | (0.21/0.26) | (85.49/83.70) | (4.13/6.19) | (428/617) |
| MinAlt | (7.33/9.19) | (0.59/0.95) | (1.50/3.38) | (0.57/1.01) | (1.10/-2.08) | (0.60/0.97) | (21.73/9.74) | (0.14/0.21) | (75.00/59.16) | (2.05/2.70) | (491/525) |

## 3 Results and Discussion

### 3.1 Case Studies

Two case studies are first presented to demonstrate how the co-location and comparisons were done between the dropsondes and the Falcon instrumentation. Research Flight 25 on 20 August 2020 was a case of measurements being compared in cloud-free conditions at the co-located time identified using the method in Figure 1. Figure 4a shows the aircraft flight tracks overlaid on GOES 16 visible imagery (15:41 UTC), Figure 4b shows a time series of both the Falcon flight altitude and dropsonde altitude, while Figures 4c-d zoom in on the ~30 min where the final points were selected along with time series of TAMMS wind speed/wind direction/temperature and DLH measurements of relative humidity. Figure 4e shows dropsonde vertical profiles of temperature and relative humidity for this case. The Falcon was conducting its routine stairstep flight profile at various levels below, in, and above clouds on this particular day and the final Falcon (red marker) and dropsonde (green marker) data points identified as a matching pair were coincidentally both out of cloud at their respective times. The altitudes of the chosen Falcon and dropsonde points were 502.8 m and 502.5 m, respectively, with a horizontal separation distance between them of 3.1 km and a time difference of 14 minutes and 2 seconds. The temperature time series expectedly shows

clear differences at the various Falcon legs at the different altitudes with higher temperatures at the lower altitude legs; this helps support the choice of trying to keep the vertical separation distance narrow in the methodology (Fig. 1) even though it comes at expense of greater time and horizontal separation. Relative humidity from the Falcon was highly variable along this time segment, ranging from as low as 51.8% to as high as 98.7%. The lower values are due to the early portion of the time segment shown (Fig. 4c-e) when the aircraft was above cloud tops where there were drier conditions. The wind speed time

series show variations from as low as 4.0 m s$^{-1}$ to as high as 9.7 m s$^{-1}$ during the 30 min time segment shown. The final pair of points chosen yielded the following wind speed, wind direction, temperature, and relative humidity values (in that order): Falcon = 6.6 m s$^{-1}$, 242.6°, 20.6 °C, 75.1%; dropsonde = 5.1 m s$^{-1}$, 244.2°, 21.9°C, 56.4%. This case shows that the co-location method is prone to limitations due to the horizontal separation distance where atmospheric conditions can vary due to spatial gradients, especially for relative humidity in areas of considerable spatial gradients with regards to cloud top and marine

boundary layer heights (Xu et al., 2024); however, still the intercomparison shows reasonable agreement at least for wind speed/direction and temperature.

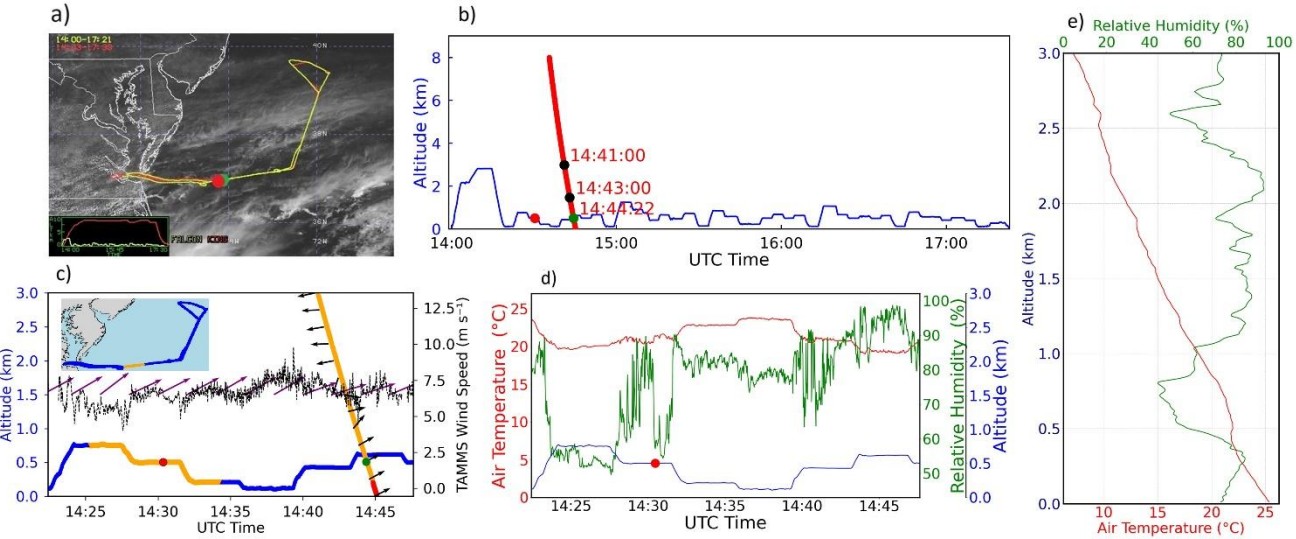

**Figure 4: Visualization of the Falcon aircraft's flight path for Research Flight 25 on 20 August 2020 along with**

**additional panels to demonstrate the methodology of co-location following the approach shown in Fig. 1 with cloud-free conditions at the point of data point selection for the dropsonde (green marker in all panels) and the Falcon instruments (red marker in all panels). (a) Flight tracks of the King Air (red) and the Falcon (yellow) overlaid on GOES-16 imagery at 15:41 UTC. (b) Time series of the Falcon altitude and the dropsonde altitude. c) Close-up of panel (b) between 14:22 UTC and 14:47 UTC focusing on where the final points were identified as a dropsonde-Falcon pair**

**(red = TAMMS point at 14:30:21, green = dropsonde point at 14:44:23). Falcon altitude is on left y-axis and its colors**

**(blue and orange) match the colors on the inset map to show the location of the orange area that includes the Falcon data within 30 km horizontal separation and 15 min temporal separation of the falling dropsonde. TAMMS wind speed is on the right y-axis and its wind direction measurements are shown as purple arrows. For the dropsonde profile (shown as the steep diagonal line), wind directions are shown as black arrows and the orange portion indicates that**
**those points are within the 30 km horizontal separation and 15 min separation criteria (i.e., red parts of the diagonal line at bottom violate the criteria). (d) Time series of the Falcon's temperature and relative humidity measurements along the same time segment as panel (c) with the red colored marker again showing the time of the final selected TAMMS point for this identified pair. (e) Dropsonde vertical profile of temperature and relative humidity.**

The second example from Research Flight 33 on 11 September 2020 was a case such that the final dropsonde and Falcon points were in cloud (Fig. 5a-e). The altitudes of the chosen Falcon and dropsonde points were 320.3 m and 319.6 m, respectively, with a horizontal separation distance of 3.7 km and a time difference of 8 minutes and 30 seconds. The final pair of points chosen yielded the following wind speed, wind direction, temperature, and relative humidity values (in that order): Falcon = 6.6 m s$^{-1}$, 61.8°, 24.6°C, 91.8%; dropsonde = 5.7 m s$^{-1}$, 79.4°, 24.3°C, 97.7%.

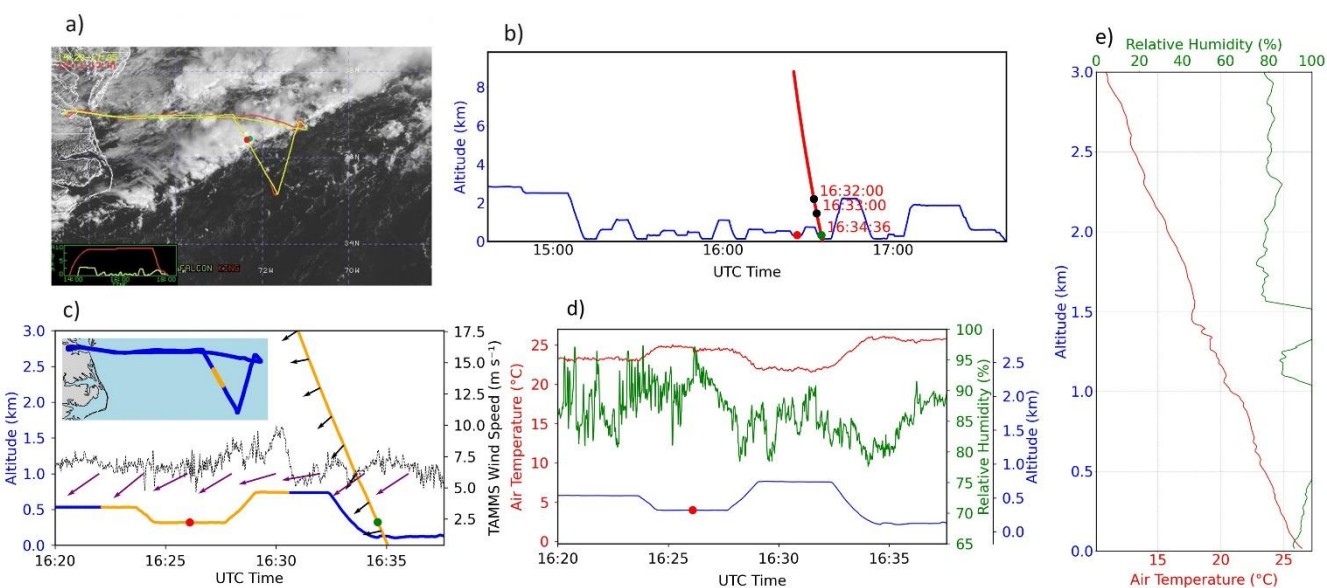


**Figure 5: Same as Fig. 4 but for a Research Flight 33 on 11 September 2020, coinciding with the final chosen points being in cloud.**

## 3.2 TAMMS – Dropsonde Wind Variable Intercomparisons

Figure 6 intercompares wind speeds from the TAMMS and dropsondes. TAMMS-dropsonde data pairs (369,677 pairs) are
first examined using the method shown in Figure 1 excluding the last step of finding just one pair of points for a dropsonde launch with the minimum altitude difference. In other words, all pairs are allowed if vertical separation is within 25 m, while

keeping other criteria still in place (i.e., horizontal separation < 30 km, time separation < 15 min, Falcon roll angle < 5°, and clear/cloud criteria). After applying the final step of Figure 1's criteria, the resultant 555 pairs exhibit a range of wind speeds from (TAMMS) 0.30 m s$^{-1}$ to 40.83 m s$^{-1}$ and (dropsonde) 0.17 m s$^{-1}$ to 42.16 m s$^{-1}$, as shown by histograms in Figure 7a-b, which report various statistics (mean, standard deviation, 25/33/50 (median)/67/75 percentiles). The median wind speeds (TAMMS/dropsonde) were 7.64/7.23 m s$^{-1}$.

The intercomparison results are comparable between relaxed (Figure 6a) and stricter vertical separation criteria (Figure 6b). Results in Figure 6a show that the slopes are 0.97 and 1.02 and y-intercepts are 0.43 and 0.01 for the linear and bisector regressions, respectively, while the correlation coefficient was 0.95. For the bisector regression results, which assume both variables are subject to error, the slope and y-intercept demonstrate excellent agreement between the two datasets. Figure 6b shows slopes of 0.98 and 1.03 and y-intercepts of 0.39 and -0.08 for the linear and bisector regressions, respectively, with a correlation coefficient of 0.95. The mean errors (± STD) corresponding to Fig. 6a and 6b are 0.21 ± 1.62 m s$^{-1}$ and 0.21 ± 1.68 m s$^{-1}$, respectively (Table 3). Therefore, the TAMMS had slightly higher wind speed values on average. This is assuring that the stricter criteria of using just one TAMMS-dropsonde pair per dropsonde launch by applying the full method in Fig. 1 is representative of more loose criteria that would provide more data points. These two methods of intercomparisons with vastly different datapoints were conducted for the other geophysical variables of interest in this study (u/v wind components, wind direction, temperature, relative humidity) and the same general conclusion is reached that the final intercomparison metrics are similar between the two approaches.

To further investigate wind speed intercomparisons, Table 3 relies on the final 555 pairs and compares results between seasons (winter versus summer), cloudiness (clear vs cloud), and different terciles of (i) overall wind speed, (ii) altitude at which the final pairs were identified, and (iii) horizontal distance between the two data points in each pair. The terciles were meant to keep the data points the same for three categories (185) being compared using the 33$^{rd}$ and 67$^{th}$ percentile values reported in Figure 7. The summer and winter data intercomparisons show similar correlation coefficients (0.94 for both), mean errors (0.17 ± 1.29 m s$^{-1}$ = summer, 0.27 ± 2.02 m s$^{-1}$ = winter), and slopes (0.97-1.05 depending on season and regression type). There also was no major difference in the agreement when using pairs in cloud versus no cloud with correlation coefficients being between 0.89 (cloudy) and 0.95 (clear), mean errors between -0.13 ± 1.98 m s$^{-1}$ (cloudy) and 0.26 ± 1.60 m s$^{-1}$ (cloudy), and slopes between 0.87 and 1.04.

Intercomparisons of wind speed for terciles of altitude and horizontal separation distances revealed generally good agreement with the six possible slopes shown for these different categories and regression types being between 0.92 and 1.06 with correlation coefficients between 0.92 and 0.96. The mean error though increased for the highest altitude tercile (> 1304 m), with a value of 0.58 ± 1.82 m s$^{-1}$ as compared to mean errors between -0.01 m s$^{-1}$ and 0.07 m s$^{-1}$ for the lower two altitude tercile categories. It was hypothesized that the agreement should be best for the lowest horizontal separation distances. This is partly confirmed since the highest separation tercile (> 22,955 m) exhibited the highest mean error (0.45 ± 1.84 m s$^{-1}$) as compared to the lowest two terciles that had mean errors of 0.06 to 0.13 m s$^{-1}$. The mean errors for the three wind speed terciles (< 5.80 m s$^{-1}$, 5.80-9.71 m s$^{-1}$, > 9.71 m s$^{-1}$) were between 0.09 m s$^{-1}$ and 0.41 m s$^{-1}$ with more variability in values for slopes,

y-intercepts, and correlation coefficients compared to the other comparison factors in Table 3. For instance, based on the bisector method, which assumes error in both datasets, the slopes and y-intercepts were much more improved for the low (1.37 and -1.05 m s$^{-1}$, respectively) and high wind speed categories (1.14 and -1.81 m s$^{-1}$, respectively) as compared to the medium speed category (2.28 and -9.50 m s$^{-1}$, respectively). Correlation coefficients ranged widely from 0.55 for the medium wind tercile to as high as 0.90 for the high wind speed tercile. The standard deviation in the mean errors increased as a function of wind speed, but if divided by the mean wind speed in each tercile, the standard deviations are similar.

Lastly, Table 3 additionally summarizes intercomparison metrics for TAMMS-dropsonde pairs but with the inclusion of 5, 11, and 21 data points for TAMMS rather than 1, and 3 points for dropsondes rather than 1 (all sets of points are centered by the single point used for the previous discussion). The results show very similar results with mean errors between 0.21 and 0.23 m s$^{-1}$, correlation coefficients of 0.94-0.95 for all three types of comparisons, and slopes being between 0.94 and 0.96. In terms of wind's two horizontal components, Tables S1-S2 show improved results in terms of how well different categories compare when contrasted with Table 2 for total wind speed.

Another type of analysis shown on the right hand side of Fig. 7 for all compared variables is the frequency histogram of the difference in values between the Falcon and dropsonde measurements. For wind speed and u and v components, the median difference shown was 0.18, 0.09, and -0.03 m s$^{-1}$, indicative of the lack of a substantial bias between the TAMMS and dropsonde measurements.

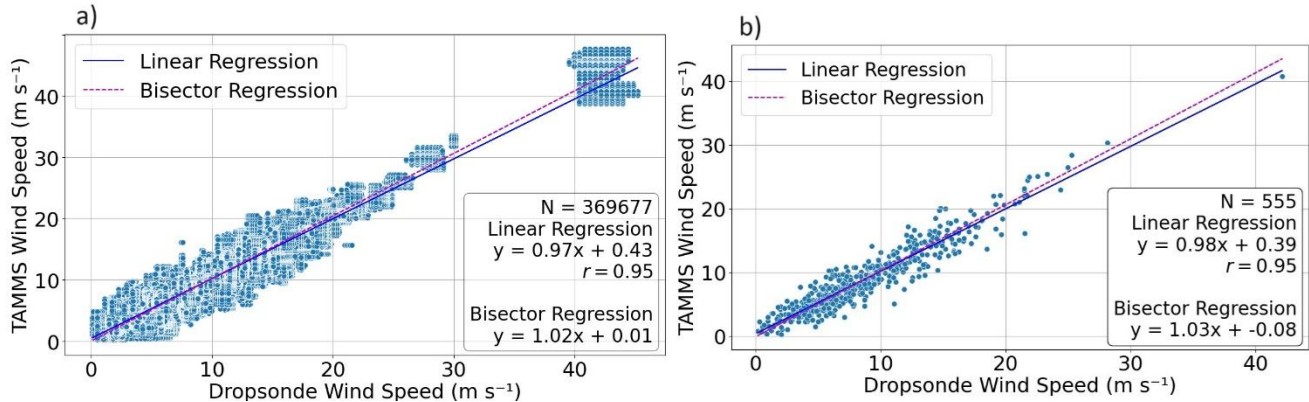

**Figure 6: (a) Wind speed scatterplot between dropsonde and TAMMS pairs of single points following the method of Figure 1 (within 30 km, 15 min, and 25 m vertical spacing) excluding the last step to find the single points with the minimum altitude difference. (b) Same as (a) except showing only pairs following the full method of Figure 1 including the last step.**

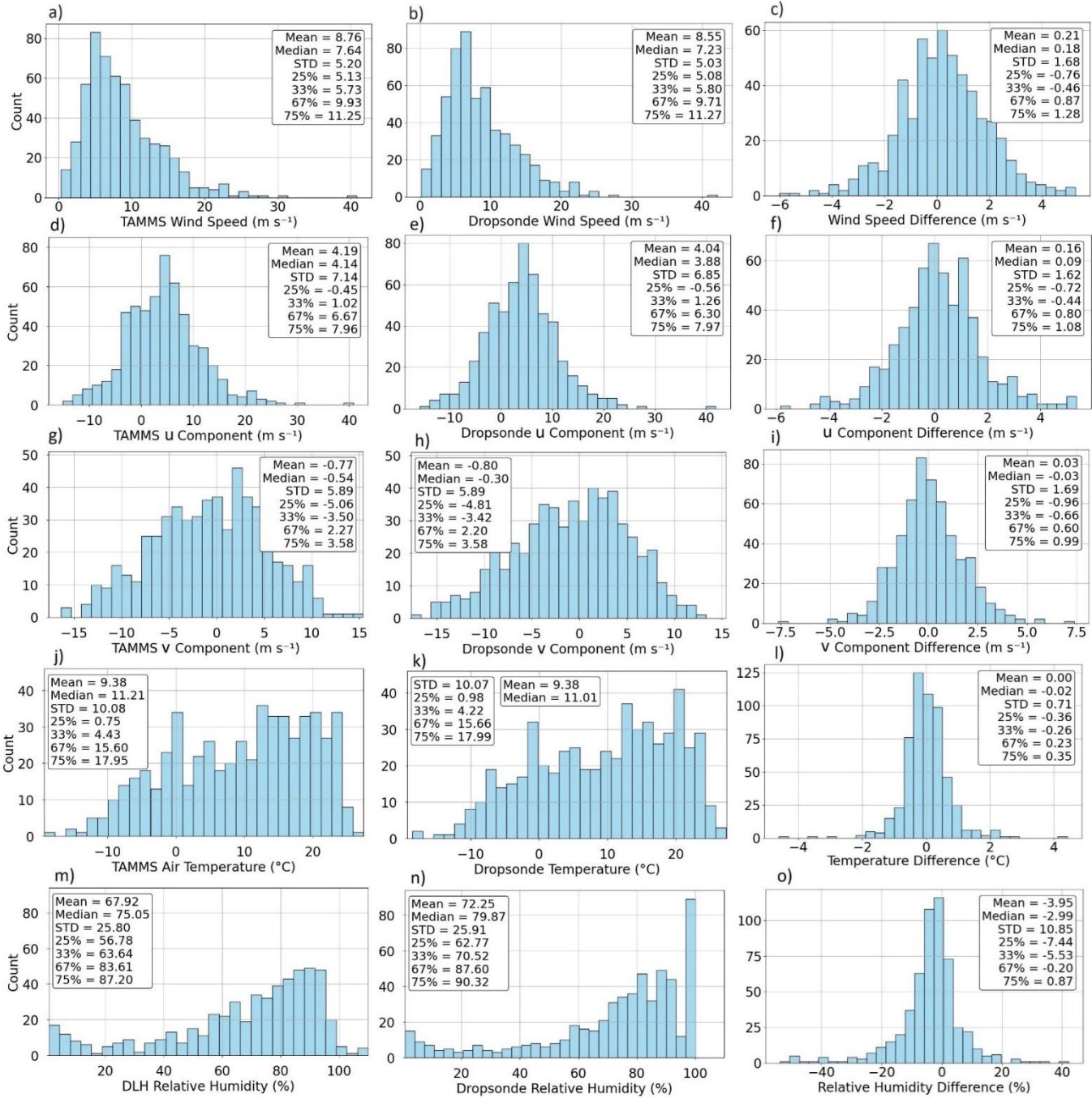

**Figure 7:** Histogram TAMMS and dropsonde (a-b) wind speed, (d-e) u component, (g-h) v component, (j-k) TAMMS and dropsonde temperature, (m-n) DLH and dropsonde relative humidity. Note that this study only uses dropsonde

**relative humidity values below 100%. Panels c/f/i/l/o show histograms of the differences of each variable calculated as such: Falcon measurements (i.e., TAMMS or DLH) minus dropsonde measurements.**

As it relates to wind direction, the overall wind rose for the final 555 pairs (based on single points) is shown in Figure 8a and
8b for the TAMMS and dropsondes, respectively. Both wind rose plots show similarity with wind direction predominantly being from the northwest but extending usually to southwesterly, as is expected based on the climatology of the region and documented atmospheric circulation patterns (Sorooshian et al., 2020; Painemal et al., 2021). As quantitative intercomparisons for wind direction are complicated owing to its native values not being conducive to such an analysis (e.g., 0° and 360° essentially being the same), Figure 8c intercompares the cosine of the respective wind directions with generally good
agreement between the two datasets (R = 0.94, linear/bisector regression slopes of 0.94/1.00 and mean error of 0.00 ± 0.22). Markers are colored by TAMMS wind speed to see if deviations in wind direction have any dependence on speed but the results do not show such a dependence.

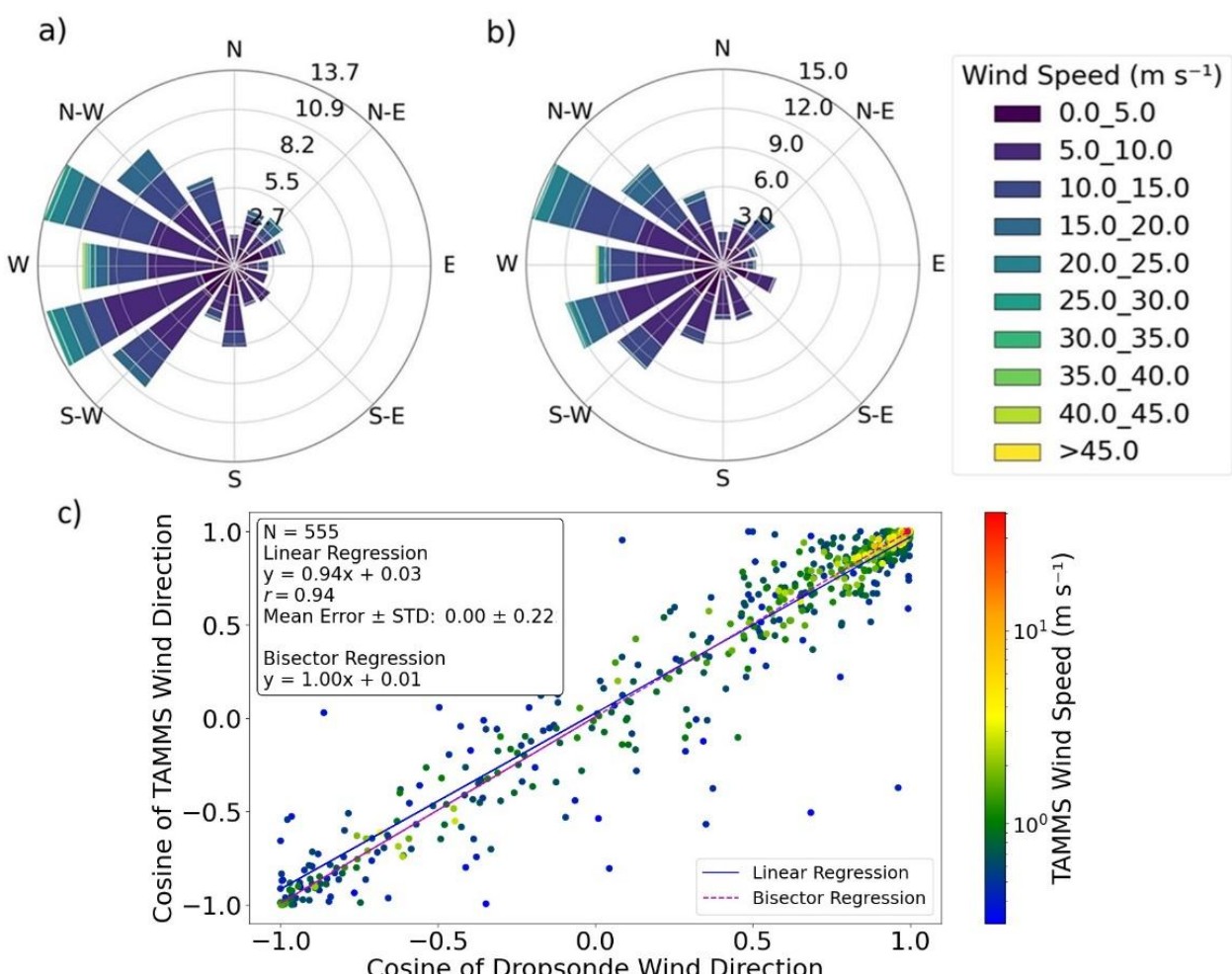

**Figure 8: Wind rose for the final 555 pairs of points for a) TAMMS and b) dropsonde. c) Wind direction scatterplot**
**between dropsonde and TAMMS pairs of single points following the full method of Figure 1 including the last step.**
**Note that the cosine of wind direction is used to handle values close to 0/360° in a more meaningful way. Markers are**
**colored by TAMMS wind speed on a logarithmic scale.**

### 3.3 Intercomparison of Temperature and Humidity Measurements

Following the same format of the wind speed results in Figure 6, Figure 9 intercompares temperature and relative humidity measurements between the Falcon and dropsondes. Statistics associated with the intercomparisons for temperature and relative humidity are in Table 4 and 5, respectively. Histograms of these two variables (Figure 7i-l) reveal median values of temperature (Falcon/Dropsonde) of 11.21/11.01 °C with a minimum of -19.24/-18.47 °C and maximum of 27.36/27.30 °C. Median values of relative humidity (Falcon/Dropsonde) are 75.05/79.87% with a minimum of 0.70/0.72% and maximum of 109.90/99.99%.

The overall correlation coefficients are 0.99 for temperature when using all data pairs with relaxed vertical gap criteria (369,859 pairs). For the final 555 pairs, temperature intercomparisons revealed almost perfect results with a mean error of 0.00 ± 0.71 °C, correlation coefficient of 0.99, and slopes of 1.00 for both linear and bisector regressions. Y-intercepts were between -0.01 and 0.01 °C. For all the various other categories examined, temperature results generally remained nearly perfect in terms of agreement with low mean errors (between -0.13 and 0.12 °C) and correlation coefficients ≥ 0.98. The median difference in temperature (TAMMS – dropsonde) was -0.02 °C (Fig. 7l).

Relative humidity values did not agree as well as temperature due at least partly to more general spatial heterogeneity that exists with relative humidity in the atmosphere as confirmed with previous studies using the ACTIVATE data (e.g., Crosbie et al., 2024; Dadashazar et al., 2022). Figure 9c-d shows relative humidity scatterplots colored by the temperature difference between TAMMS and dropsondes because the DLH relative humidity values are a derived quantity and depend on temperature; thus it is of interest to identify possible sensitivity of the relative humidity instrument differences to a temperature bias. Figure 9c-d indeed does show that some of the larger relative humidity differences exhibit the largest temperature biases too. For the first two categories compared in Table 5, the mean error was (relaxed vertical criteria) -3.08 ± 11.22% and (strict vertical criteria) -3.86 ± 10.74%, with correlation coefficients being 0.93 and 0.91 for those two categories, respectively. Thus it seems that the dropsondes usually measured higher relative humidity on average, which is a consistent theme throughout Table 5 when examining the mean error for the various categories shown. Figure 7o shows that the median difference between the DLH and dropsonde relative humidity was -2.99%, further confirming a slight bias with higher dropsonde values. The relative humidity agreement appeared to be best when examining the lowest relative humidity tercile (< 70.52%) as compared to any other category shown, which indicates that this comparison is not favorable in more humid conditions such as near clouds with substantial spatial variability. This was already demonstrated in Figure 4's case study flight. For that lowest tercile of relative humidity conditions, the mean error is -0.30 ± 10.16% with a correlation coefficient of 0.91 and linear/bisector slopes of 0.98/1.08. Still even for that category, the standard deviation was considerable.

Given that relative humidity is temperature-dependent, water vapor mixing ratio is also examined. With relaxed vertical criteria, the correlation between DLH and dropsonde water vapor mixing ratio values remained strong (r = 0.97; Fig. 9e), similar to the relative humidity intercomparisons (r = 0.93). With stricter vertical criteria (Fig. 9f), the intercomparison was still strong (r = 0.84) and consistent with the relative humidity intercomparisons (r = 0.91). This further confirms the robustness of the humidity measurements across different instruments, regardless of temperature differences.

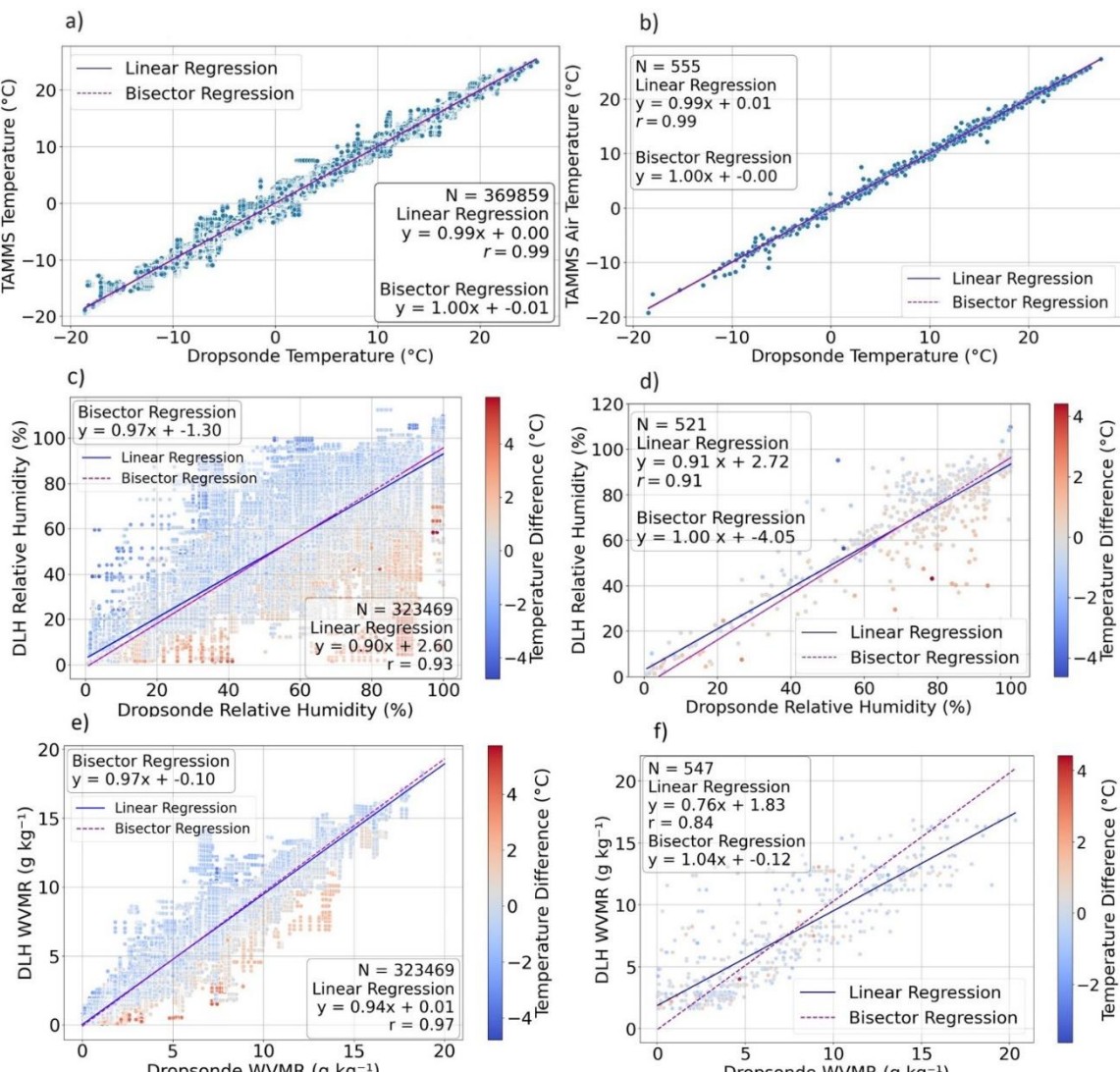

**Figure 9:** (a) **Temperature scatterplot between dropsonde and TAMMS pairs of single points following the method of Figure 1 (within 30 km, 15 min, and 25 m vertical spacing) excluding the last step to find the single points with the minimum altitude difference. (b) Same as (a) except showing only pairs following the full method of Figure 1 including the last step. (c-d) Same as (a-b), but for relative humidity with markers colored by temperature difference: TAMMS – dropsonde. (e-f) Same as c-d, but for water vapor mixing ratio (WVMR).**

**Table 3: Summary table of all TAMMS – dropsonde wind speed intercomparison metrics. The first row uses all data points following the Fig. 1 method excluding the last step, which allows for vastly increased statistics by allowing all pairs to be used that were within 25 m vertical distance. The second row includes the final step of Fig. 1 using only the**

data points with minimized vertical separation (i.e., one pair per dropsonde launch). Remaining rows examine all the data from the second row but in various categories. SE = standard error; STD = standard deviation.

| | N | Linear (SE)/Bisector Slope (SE) | Linear /Bisector Y-Intercept (m s$^{-1}$) | r | Mean Error ± STD (m s$^{-1}$) |
|---|---|---|---|---|---|
| All: Relaxed vertical criteria | 369677 | 0.97 (0.00)/1.02 (0.00) | 0.43 /0.01 | 0.95 | 0.21 ± 1.62 |
| All: Strict vertical criteria | 555 | 0.98 (0.01)/1.03 (0.02) | 0.39/-0.08 | 0.95 | 0.21 ± 1.68 |
| Summer | 293 | 0.98 (0.02)/1.05 (0.03) | 0.29/-0.17 | 0.94 | 0.17 ± 1.29 |
| Winter | 262 | 0.97 (0.02)/1.03 (0.03) | 0.61/-0.05 | 0.94 | 0.27 ± 2.02 |
| Cloudy | 81 | 0.87 (0.06)/0.98 (0.09) | 1.11/0.06 | 0.89 | -0.13 ± 1.98 |
| Clear | 465 | 0.99 (0.01)/1.04 (0.02) | 0.35/-0.07 | 0.95 | 0.26 ± 1.60 |
| WS ≤ 5.80 m s$^{-1}$ | 185 | 0.83 (0.07)/1.37 (0.09) | 1.08/-1.05 | 0.67 | 0.41 ± 1.30 |
| 5.80 < WS ≤ 9.71 m s$^{-1}$ | 185 | 0.89 (0.10)/2.28 (0.21) | 0.91/-9.50 | 0.55 | 0.09 ± 1.57 |
| WS > 9.71 m s$^{-1}$ | 185 | 1.01 (0.04)/1.14 (0.04) | 0.03/-1.81 | 0.90 | 0.14 ± 2.04 |
| Altitude ≤ 673 m | 185 | 0.95 (0.03)/1.01 (0.03) | 0.49/0.01 | 0.94 | 0.07 ± 1.46 |
| 673 < Altitude ≤ 1304 m | 185 | 0.92 (0.03)/1.00 (0.03) | 0.67/-0.04 | 0.92 | -0.01 ± 1.63 |
| Altitude > 1304 m | 185 | 1.01 (0.02)/1.05 (0.02) | 0.52/0.12 | 0.96 | 0.58 ± 1.82 |
| Horiz. distance ≤ 13616 m | 185 | 0.95 (0.02)/1.01 (0.02) | 0.51/0.08 | 0.95 | 0.13 ± 1.41 |
| 13616 < Horiz. dist. ≤ 22955 m | 185 | 0.96 (0.03)/1.02 (0.03) | 0.45/-0.13 | 0.94 | 0.06 ± 1.70 |
| Horiz. distance > 22955 m | 185 | 1.01 (0.02)/1.06 (0.02) | 0.39/-0.07 | 0.95 | 0.45 ± 1.84 |
| 5 TAMMS Pts/3 Dropsonde Pts | 485 | 0.96 (0.01)/0.94 (0.02) | 0.54/0.64 | 0.94 | 0.21 ± 1.61 |
| 11 TAMMS Pts/3 Dropsonde Pts | 467 | 0.95 (0.01)/0.95 (0.02) | 0.59/0.63 | 0.95 | 0.23 ± 1.55 |
| 21 TAMMS Pts/3 Dropsonde Pts | 448 | 0.96 (0.01)/0.95 (0.02) | 0.56/0.62 | 0.95 | 0.23 ± 1.51 |


**Table 4: Same as Table 3 but for temperature.**

| | N | Linear (SE)/Bisector Slope (SE) | Linear/Bisector Y-Intercept (°C) | r | Mean Error ± STD (°C) |
|---|---|---|---|---|---|
| All: Relaxed vertical criteria | 369859 | 0.99 (0.00)/1.00 (0.00) | 0.00 /-0.01 | 0.99 | 0.00 ± 0.73 |
| All: Strict vertical criteria | 555 | 1.00 (0.00)/1.00 (0.00) | 0.01/-0.01 | 0.99 | 0.00 ± 0.71 |
| Summer | 293 | 1.00 (0.01)/1.00 (0.01) | -0.02/-0.11 | 0.99 | -0.03 ± 0.67 |
| Winter | 262 | 1.00 (0.01)/1.01 (0.01) | 0.03/0.02 | 0.99 | 0.04 ± 0.76 |
| Cloudy | 81 | 1.01 (0.01)/1.01 (0.01) | -0.22/-0.23 | 0.99 | -0.18 ± 0.68 |
| Clear | 465 | 0.99 (0.00)/1.00 (0.00) | 0.10/0.07 | 0.99 | 0.03 ± 0.70 |
| Temperature ≤ 4.22 °C | 185 | 1.01 (0.01)/1.03 (0.02) | 0.02/0.06 | 0.98 | -0.02 ± 0.89 |
| 4.22 < Temperature ≤ 15.66 °C | 185 | 1.00 (0.01)/1.02 (0.02) | 0.09/-0.10 | 0.98 | 0.12 ± 0.64 |
| Temperature > 15.66 °C | 185 | 0.99 (0.01)/1.01 (0.02) | 0.18/-0.22 | 0.98 | -0.10 ± 0.54 |
| Altitude ≤ 673 m | 185 | 0.99 (0.00)/0.99 (0.01) | 0.23/0.20 | 0.99 | 0.02 ± 0.51 |
| 673 < Altitude ≤ 1304 m | 185 | 1.00 (0.00)/1.00 (0.01) | -0.03/-0.04 | 0.99 | -0.04 ± 0.55 |
| Altitude > 1304 m | 185 | 1.01 (0.01)/1.01 (0.01) | -0.00/-0.03 | 0.99 | 0.02 ± 0.97 |
| Horiz. distance ≤ 13616 m | 185 | 0.99 (0.00)/1.00 (0.01) | 0.02/0.01 | 0.99 | -0.03 ± 0.57 |
| 13616 < Horiz. dist. ≤ 22955 m | 185 | 1.00 (0.00)/1.00 (0.01) | -0.04/-0.06 | 0.99 | -0.03 ± 0.67 |
| Horiz. distance > 22955 m | 185 | 1.00 (0.01)/1.01 (0.01) | 0.05/0.02 | 0.99 | 0.07 ± 0.86 |
| 5 TAMMS Pts/3 Dropsonde Pts | 514 | 0.99 (0.00)/0.99 (0.00) | 0.01/0.01 | 0.99 | 0.00 ± 0.67 |
| 11 TAMMS Pts/3 Dropsonde Pts | 514 | 0.99 (0.00)/0.99 (0.00) | 0.01/0.00 | 0.99 | -0.01 ± 0.66 |
| 21 TAMMS Pts/3 Dropsonde Pts | 514 | 0.99 (0.00)/0.99 (0.00) | 0.00/0.00 | 0.99 | -0.01 ± 0.64 |

**Table 5: Same as Table 3 but for relative humidity.**

| | N | Linear (SE)/Bisector Slope (SE) | Linear /Bisector Y-Intercept (%) | r | Mean Error ± STD (%) |
|---|---|---|---|---|---|
| All: Relaxed vertical criteria | 323469 | 0.90 (0.00)/0.97 (0.00) | 2.60 /-1.30 | 0.93 | -3.08 ± 11.22 |
| All: Strict vertical criteria | 521 | 0.91 (0.02)/1.00 (0.03) | 2.72/-3.73 | 0.91 | -3.86 ± 10.74 |
| Summer | 286 | 0.86 (0.03)/1.02 (0.04) | 6.36/-4.92 | 0.85 | -3.76 ± 10.94 |
| Winter | 235 | 0.92 (0.02)/0.99 (0.03) | 1.10/-3.18 | 0.93 | -3.99 ± 11.09 |
| Cloudy | 56 | 0.62 (0.73)/5.40 (0.74) | 33.61/-438.80 | 0.11 | -4.22 ± 5.38 |
| Clear | 465 | 0.89 (0.02)/1.00 (0.03) | 3.36/-3.64 | 0.90 | -3.82 ± 11.50 |
| RH ≤ 70.52% | 174 | 0.98 (0.03)/1.08 (0.05) | 0.37/-3.61 | 0.91 | -0.30 ± 10.16 |
| 70.52% < RH ≤ 87.60% | 174 | 0.62 (0.20)/2.73 (0.22) | 24.88/-139.81 | 0.23 | -4.61 ± 11.99 |
| RH > 87.60% | 173 | 1.22 (0.16)/2.40 (0.17) | -26.74/-136.97 | 0.51 | -6.68 ± 9.55 |
| Altitude ≤ 673 m | 174 | 0.88 (0.04)/1.04 (0.06) | 4.64/-8.36 | 0.84 | -4.86 ± 8.94 |
| 673 < Altitude ≤ 1304 m | 174 | 0.90 (0.05)/1.08 (0.06) | 4.18/-9.75 | 0.83 | -3.90 ± 10.80 |
| Altitude > 1304 m | 173 | 0.91 (0.03)/0.98 (0.04) | 2.22/-1.82 | 0.93 | -2.81 ± 12.21 |
| Horiz. distance ≤ 13616 m | 174 | 0.90 (0.03)/0.97 (0.04) | 4.28/-0.86 | 0.93 | -2.82 ± 9.39 |
| 13616 < Horiz. dist. ≤ 22955 m | 174 | 0.93 (0.04)/1.04 (0.05) | 1.07/-7.28 | 0.89 | -4.34 ± 10.91 |
| Horiz. distance > 22955 m | 173 | 0.89 (0.03)/0.99 (0.05) | 2.71/-3.55 | 0.90 | -4.43 ± 11.69 |
| 5 TAMMS Pts/3 Dropsonde Pts | 468 | 0.91 (0.01)/0.92 (0.02) | 2.94/2.039 | 0.92 | -3.34 ± 9.54 |
| 11 TAMMS Pts/3 Dropsonde Pts | 467 | 0.91 (0.01)/0.92 (0.02) | 3.02/1.99 | 0.92 | -3.29 ± 9.45 |
| 21 TAMMS Pts/3 Dropsonde Pts | 464 | 0.91(0.01) /0.92 (0.02) | 3.15/2.03 | 0.92 | -3.11 ± 9.32 |


### 3.4 Intercomparison to Literature

This study's results are comparable to what is reported in the literature (Table 1) although it is not a direct comparison between studies owing to the various filtering methods employed, conditions incorporated for intercomparisons (e.g., cloudiness, altitudinal preferences, land versus ocean), and the widely ranging amount of data used between studies. For context, Dmitrovic

et al. (2024) reported a HSRL-2 wind speed accuracy of 0.15 ± 1.80 m s$^{-1}$ compared to dropsondes during ACTIVATE when focusing on ocean surface wind speed only, which is comparable to the intercomparison between dropsondes and the TAMMS in this study. When comparing air motion systems (including a TAMMS) on aircraft next to each other in flight, Thornhill et al. (2003) showed that u/v wind components and temperature intercomparisons during carefully chosen flight legs within 200 m horizontal separation exhibited slopes and y-intercepts of 1 and 0, respectively, which is similar to values in this work. Wind

direction intercomparisons by Bucci et al. (2018) between a lidar and dropsondes revealed a correlation coefficient of 0.92, while Scicluna et al. (2023) reported a r$^2$ value of 0.95 when comparing wind directions between a remotely piloted aircraft system and a ground-based lidar, with their comparisons limited to only the lowest 100 m. Those wind direction results are comparable to this study (r = 0.94) albeit this study's comparison is based on cosine of angles.

## 4. Conclusions

This study provides a novel intercomparison of wind components, temperature, and water vapor measurements by comparing data from 555 dropsondes launched from a high-flying King Air to independent in situ instruments on a lower-flying Falcon during the 2020-2022 ACTIVATE campaign over the northwest Atlantic. This effort is largely motivated by the importance of assessing airborne instrument performance and determining the validity of integrating dropsonde data with data collected from another coordinated aircraft, especially in light of a growing number of field campaigns with spatially coordinated

aircraft. ACTIVATE's large set of statistics was leveraged to compare the level of agreement for these geophysical variables in various categories related to season, cloudiness, altitude of comparison, horizontal separation distance, terciles of the variable values, and other categories associated with the amount of data points used.

The overall results reveal good agreement, with the best being for temperature (r = 0.99, mean error = 0.00 ± 0.71°C) and the weakest being for relative humidity (r = 0.91, mean error = -3.86 ± 10.74%). Overall wind speed from the TAMMS agrees

very well with dropsondes (r = 0.95; mean error = -0.21 ± 1.68 m s$^{-1}$), with its u and v components showing even better agreement. The mean errors are comparable or less than the respective instrument uncertainties (i.e., ≤0.5 m s$^{-1}$ for u/v winds, ≤0.5°C, ≤15% relative humidity). Wind directions also exhibited good agreement between the TAMMS and dropsondes. No significant biases between the respective instruments compared were noted except a small bias with slightly larger relative humidity values with dropsondes relative to the DLH, although it is shown that some of the differences can be explained by

higher temperature values measured by the TAMMS, which are used to derive relative humidity values with the DLH. This work provides added confidence in data quality for the instruments compared and highlights the suitability of using dropsonde data launched from a higher flying aircraft to another one flying lower with a fair degree of spatial and temporal coordination for conditions such as the northwest Atlantic where there can be heterogeneity in atmospheric conditions.

## Data Availability

Data from the ACTIVATE airborne campaign can be accessed at https://asdc.larc.nasa.gov/project/ACTIVATE (ACTIVATE Science Team, 2020).

## Author Contribution

EC, YC, JPD, GSD, RAF, JWH, SK, JBN, KLT, CV, HV contributed to the collection of the aircraft data. SN and SD conducted data analysis and investigation. SN and AS drafted the manuscript. All co-authors participated in the review and
editing process.

## Competing Interests

The authors declare that they have no conflict of interest.

**Disclaimer**

Publisher's note: Copernicus Publications remains neutral with regard to jurisdictional claims in published maps and
institutional affiliations.

**Acknowledgements**

The work was supported by ACTIVATE, an Earth Venture Suborbital-3 (EVS-3) investigation funded by NASA's Earth
Science Division and administered by the Earth System Science Pathfinder Program Office.

**Financial support**

University of Arizona investigators acknowledge NASA grant no. 80NSSC19K0442. CV and SK's work was supported by the
DFG SPP-1294 HALO under project no. 522359172, and by the European Union's Horizon Europe program within the Single
European Sky ATM Research 3 Joint Undertaking through the CONCERTO (grant no 101114785) and CICONIA (grant no
101114613) projects.

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
