# Peer review of "Intercomparison of wind speed, temperature, and humidity data between dropsondes and aircraft in situ measurements"

_EGUsphere, 2024_

## Author Response (AR1)

We thank the reviewers for the thorough review of our manuscript. We have addressed each of specific comments below and revised the manuscript accordingly.

**Response to Reviewer Comments**

**Reviewer 1:**

1: The article is about validating airborne in-situ measurements by dropsondes. Therefore, it is not helpful to discuss comparison between remote sensing/lidar and dropsondes, as done in the introduction and the conclusions, with a large literature review. Please either include lidar measurements in the analyses or omit it also in the introduction and conclusion. Also, the literature review in the tables should have the focus on the topic of your article, not lidar/dropsonde comparison.

Response 1:

Thank you for your feedback regarding the alignment of the manuscript's focus with its content. We included lidar/dropsonde comparisons in the introduction and literature review for several reasons. Lidar is a widely used method for validating dropsonde and in-situ measurements. Discussing these comparisons provides context for how various data sources have been validated historically and helps frame the purpose of our work. Additionally, we wanted to highlight the importance of these methods in assessing data quality, identifying potential biases, and understanding the accuracy of different data sources. Readers often prefer the bigger picture and context, and that is why we hope to convince this reviewer that the broader literature review is important to keep.

Also, highlighting lidar/dropsonde comparisons gives readers an understanding of the accuracy and challenges associated with integrating data from different sources. This broader perspective is valuable for interpreting the significance of our results, as our method introduces a novel approach to intercomparison.

To address the reviewer's concern while preserving the rationale behind including lidar comparisons, we have the following text in the Introduction:

"To provide context for this study, Table 1 provides a summary of various intercomparison studies, highlighting the methods used to validate wind speed/direction, temperature, and humidity measurements. While the primary focus of this study is on validating airborne in-situ measurements using dropsondes, intercomparisons based on other techniques are included in the table to provide important historical context, demonstrating how different datasets have been integrated and validated across platforms."

Lastly, we should add that this manuscript is not ideal for adding HSRL-2 data since those data are useful for only wind speeds around 10 m above sea level and thus not appropriate for comparison with the TAMMS winds. Also, HSRL-2 winds at 10 m have already been compared to dropsondes in other work that is cited: Dmitrovic et al. (2024).

2. Please include a thorough analysis of uncertainty of dropsondes and in-situ measurements, taking into account in particular sensor response time, see e.g. Bärfuss et al., 2018, and Bärfuss et al., 2023.

Response 2:

A conservative estimate of the uncertainties of dropsonde measurements is given by the Vaisala NRD41 data sheet:

| Pressure repeatability | 0.4 hPa |
|---|---|
| Temperature repeatability | 0.1 K |
| Temperature response time | 0.5 s |
| Relative humidity repeatability | 2 % |
| Relative humidity response time | <0.3s at +20°C < 10 s at -40°C |

The repeatability refers to the statistical variability in successive calibrations with a confidence level of k=2. The response time of the temperature sensor is given at surface pressure and 6 m/s in radiosonde applications. For dropsonde applications the ventilation speed is higher and therefore, the response time likely faster.

To achieve the expected uncertainty in relative humidity, all humidity sensors were reconditioned prior to launch to remove any contaminants, that could potentially lead to dry bias issues.

The response time of the humidity sensor varies exponentially with temperature. Since all comparisons were done at temperatures between -20°C and +25°C, the time response is likely faster than 3 s for all temperatures. Due to the fast response of the sensors in this temperature regime, the time response correction was negligible.

The data stream reports a wind speed accuracy, which is determined by the GPS receiver module directly. It is a direct estimation of the wind speed uncertainty given how well the receiver algorithms can process the received satellite signals. Wind speed uncertainties are typically less than 0.4 m/s. Winds with uncertainties larger than 0.6 m/s are rejected by the data quality control routines and specifically discussed in one of our data report manuscripts (Vömel et al., 2023).

We have provided a thorough analysis of the uncertainties, including sensor response time, in a newly created "Uncertainty Analysis and Calibration Details" section in the Supplementary Material file, which we of course reference now in the main article file; our choice to place this content in the Supplement was to keep the paper more concise and less tedious to read and to give more curious readers the option to study these deeper details in another file. This new Supplement section incorporates estimates based on the Vaisala NRD41 data sheet and cites relevant literature (e.g., Bärfuss et al., 2018; Bärfuss et al., 2023). Details include uncertainty values for pressure, temperature, relative humidity, and wind speed, as well as the methods used to account for these uncertainties. We also took the opportunity in this new Supplement section to provide details about TAMMS calibrations conducted, which is relevant to subsequent comments by this reviewer. Below is a copy of Section S1 for your reference:

**S1. Uncertainty Analysis and Calibration Details**

**a. Dropsonde Data Uncertainties**

Dropsonde data from the Vaisala NRD41 sensors have the following conservative uncertainty estimates for various parameters:

-Pressure: ±0.4 hPa (repeatability)

-Temperature: ±0.1 K (repeatability), response time 0.5 s under standard conditions

-Relative Humidity (RH): ±2% (repeatability), with response times of <0.3 s at +20°C and <10 s at -40°C

The repeatability of the sensors refers to the statistical variability in successive calibrations (confidence level: k=2). Given the higher ventilation speed in dropsonde applications compared to radiosonde use, response times are likely faster than the stated values. The response time of the humidity sensor varies exponentially with temperature. Since all RH comparisons in this study were conducted within the temperature range of -20°C to +25°C, the response time correction for RH was deemed negligible.

To ensure high data quality, all humidity sensors were reconditioned prior to deployment, mitigating potential dry bias caused by contaminants. Wind speed accuracy, directly estimated by the GPS receiver module, is typically better than ±0.4 m s$^{-1}$. Data points with uncertainties exceeding ±0.6 m s$^{-1}$ were excluded as part of the quality control process (Vömel et al., 2023).

**b. HU-25 Falcon In-Situ Wind Calibration**

As ACTIVATE was the first time the HU-25 Falcon was used for measurement of flight level winds, extensive calibrations needed to be performed to precisely determine the slope and offsets for the angles of attack and sideslip, the heading offset, and the pressure defect (example results in Figure S1). To accomplish this dedicated calibration, flights were conducted during each of ACTIVATE's six deployments to make enough repeated maneuvers above the boundary layer to build up the statistics and confidence in the values obtained while also minimizing the influence of the natural variability in the atmosphere. Most of the statistics built up were from these flights as the atmosphere desired for wind measurement calibrations (cloud-free, homogenous, and above the boundary layer) are diametrically opposed to those desired for ACTIVATE science flights.

To determine the angle of attack slope and offset, speed variations were performed at multiple altitudes varying the airspeed from near minimum to near maximum holding constant at four speeds for two minutes each (Figure S1c). The coefficients for the sideslip angle were determined through crabbing the aircraft side-to-side while holding the wings level (Figure S1b). The results of these maneuvers were repeatable from flight to flight and year to year with no unexpected outliers or results. The heading offset was determined though cross-wind reverse headings repeated at a couple of altitudes each year. Even though the heading offset is not dependent on altitude, it was done at multiple altitudes for a sanity check and to help offset any unintended influence of natural variability. The pressure defect was determined via multiple along-track reverse headings from just above the boundary layer up to about 20 kft with

enough made to be able to determine the offset as a function of Mach number very well (Figure S1a). Our calibration maneuvers separated the along track and cross track reverse headings in order to minimize the influence of natural variability on the results.

[Figure]

**Figure S1. Results of the dedicated calibration flights for the TAMMS winds system on the NASA HU-25 Falcon during the ACTIVATE field campaign. Data are for multiple years and show the repeatability in the results from year-to-year. Starting from the top, a) is from the along-track reverse headings to determine the pressure defect term, b) is the results of the tailwags (crabbing) to determine the coefficients for the sideslip angle, and c) is the results from the speed variations that provide the coefficients necessary to compute the angle of attack.**

3. A common approach to calibrate wind speed for 5-hole sondes is to perform calibration manoeuvres, e.g. flying a square and checking if the wind speed and wind direction is the same for all flight directions. Please comment on this and how the data can be improved. Was this applied to your data set?

Response 3:

As noted in the response to Comment 2 above, we provide details now about the calibration work we did with the Falcon aircraft to enhance the TAMMS data quality. See the new Section S1 above in our Response 2.

4. If you discuss in situ wind measurements based on drones, please take into account the first articles about this topic, which include a thorough comparison with conventional measurement systems (tower, radiosonde, remote sensing), e.g. Martin et al., 2011

Response 4:

Table 1 has been updated to include a brief summary of key findings from earlier drone-based wind measurement studies (including the reference provided), highlighting their contributions to the field, which is helpful to give broad context. Thank you for the suggestion.

5. It is good that you make use of measurements above the ocean, to have a larger spatial homogeneity. However, please characterize the atmospheric conditions, in particular stability. Up to which altitude does the top of the marine boundary layer reach? Maybe you can show vertical profiles of temperature or potential temperature at least for your case studies?

Response 5:

Thank you for your suggestion regarding the characterization of atmospheric stability and the marine boundary layer. We added the following text to help address this comment:

"Furthermore, showcasing data across different seasons over three years naturally presents differences in atmospheric stability and boundary layer structure, which is relevant to consider in this study focused on intercomparisons of temperature, humidity, and wind speed and direction. For context, recent work relying on 506 dropsondes launched during ACTIVATE (Xu et al., 2024) revealed that the median cloud fraction for ACTIVATE flights was 0.22 with the median planetary boundary layer height for cloud fractions below and above 0.22 being 659 m and 1,169 m, respectively. Further, ~28% of those dropsondes indicated the presence of decoupled boundary layers. In light of this type of variability, steps are still taken to enhance meaningful intercomparisons between dropsondes and in situ data such as accounting for separation in time and horizontal/vertical distances along with comparing data between similar seasons and when clouds are present or not."

We also now provide vertical profiles of temperature and humidity for the two case studies.

[Figure]

Figure 4

[Figure]

Figure 5

- Xu, Y., Mitchell, B., Delgado, R., Ouyed, A., Crosbie, E., Cutler, L., et al. (2024). Boundary layer structures over the northwest Atlantic derived from airborne high spectral resolution lidar and dropsonde measurements during the ACTIVATE campaign. Journal of Geophysical Research: Atmospheres, 129, e2023JD039878. https://doi.org/10.1029/2023JD039878

6. Dropsonde data: according to l. 117, there were corrections and smoothing algorithms applied. Please summarize shortly what was done to the data. This is highly important in this context.

Response 6:

Thank you for your comment highlighting the need to clarify the corrections and smoothing algorithms applied to the dropsonde data. We added a detailed paragraph to the methods section summarizing the key steps, including dynamic corrections for sensor lag and sonde inertia, the use of a 5-second smoothing window with a B-spline algorithm, recalculation of wind parameters, and post-processing of outliers and telemetry issues. These updates ensure that the data processing steps are thoroughly described and aligned with standard procedures as outlined in Vömel et al. (2023) and Aberson et al. (2023).

"The dropsonde data underwent a series of corrections and quality control steps to ensure accuracy and reliability, following the methods detailed in Vömel et al. (2023) and Aberson et al. (2023). These included dynamic corrections to account for the lag in temperature and humidity sensors, with an equilibration time of 10 seconds applied to eliminate artifacts after release from the aircraft. Wind data were corrected for sonde inertia to address motion-induced errors during descent. A B-spline smoothing algorithm was employed for pressure, temperature, humidity, and wind components, with a 5-second smoothing window to maintain profile continuity while preserving critical gradients. Post-smoothing, wind speed and direction were recalculated to ensure consistency. Additionally, outliers and suspect data points were removed, and surface pressure values were extrapolated using fall rates, followed by recalculating geopotential height and vertical wind velocity. Any telemetry issues, such as synchronization errors between the onboard computer and GPS, were addressed through post-processing and reconstruction of raw data. These steps collectively ensured high-quality datasets suitable for robust analysis (Aberson et al., 2023; Vömel et al., 2023)."

7. RH values above 100% were set to 100%. Please explain this. Why was the obviously erroneous data not corrected or excluded? What does this mean for uncertainty?

Response 7:

Thank you for pointing out this issue. We apologize for the confusing sentence in the original manuscript. We did indeed exclude the dropsonde data for RH ≥ 100%. This is why the number of points included in the RH analysis is lower compared to other analyses (521 compared to 555).

We have carefully revised all related sections in the manuscript to clarify this and ensure the methodology is accurately described. Additionally, we acknowledge that excluding these data points helps maintain the accuracy of the analysis by removing potentially erroneous values. Here is revised part:

"Customarily, RH measurements above 100% are set equal to 100% in quality controlled data (Vömel et al., 2023). Typically, raw measurements may exceed the 100% level by a few percent due to measurement uncertainty at saturation and due to environmental factors, mostly the presence of liquid water in clouds. However, here we decided to exclude all values ≥ 100% to ensure the accuracy and reliability of the results. In doing so, we aimed to eliminate potentially erroneous measurements due to supersaturation and the potential presence of liquid water that could introduce biases into the analysis. Consequently, the number of points used for the RH analysis is lower than in the other analyses, reflecting this exclusion process. Intercomparisons among variables other than RH are still conducted if dropsonde RH ≥ 100%. Details on the uncertainty analysis for dropsondes, including sensor response times and calibration procedures, are provided in the Supplementary Material file (Text S1). This includes estimates for pressure, temperature, relative humidity, and wind speed uncertainties as supported by literature (Bärfuss et al., 2018; Bärfuss et al., 2023; Vömel et al., 2023)."

8. According to l. 135, extensive calibrations were applied. Please summarize shortly.

Response 8:

As noted in a previous response above (Comment #2-3), a detailed discussion of instrument uncertainty, calibration maneuvers, and sensor response times, along with a multi-panel figure showing results of such extensive calibrations, is now provided in the Supplementary Material.

9. 138: "wind data derived from aircraft are sensitive to deviations away from straight and level flight conditions": please quantify, and indicate how you treated this problem in the data analysis. What is "no rapid changes in altitude"? Please quantify.

Response 9:

As described in the manuscript (Section 3.2), the wind data analysis was restricted to segments where the aircraft maintained straight and level conditions, which is quantified as having pitch and roll angles less than ±5°. We clarify these details with the following text in the draft:

"As such, we restrict our data usage to times when the HU-25 is flying straight and level (pitch and roll angles less than 5 degrees (absolute))."

10. 145/146: The RH values have an uncertainty of 5%. Then the calculated RH values have a higher uncertainty? How were they calculated? Why is the uncertainty higher?

Response 10:

Thank you for highlighting this point. Relative humidity (RH) is derived from water vapor measurements obtained with a diode laser hygrometer (DLH), which has an uncertainty of 5% or 0.1 ppmv (Sorooshian et al., 2023). The calculation of RH involves additional variables, including temperature and pressure, each contributing their own uncertainties. For example, the saturation vapor pressure, which depends exponentially on temperature, is sensitive to small uncertainties in temperature measurements (e.g., ±0.2 K). Similarly, the conversion of water vapor mixing ratio into partial pressure introduces uncertainties from pressure measurements (e.g., ±1 hPa). These combined factors result in the propagated uncertainty for calculated RH values being higher than the direct water vapor measurements, depending on atmospheric conditions.

We revised the manuscript to include a detailed explanation of the calculation process and the associated uncertainties to improve clarity and transparency.

"Relative humidity (RH) is derived from water vapor measurements using a diode laser hygrometer (DLH), which has an uncertainty of 5% or 0.1 ppmv (Sorooshian et al., 2023). RH calculations also depend on temperature and pressure measurements, introducing additional uncertainties that propagate into the final values (Garcia Skabar, 2015). The saturation vapor pressure, determined by temperature, is

particularly sensitive, with small temperature uncertainties leading to noticeable RH variations. Similarly, pressure measurements contribute to the total uncertainty. As a result, the combined uncertainty of RH is higher than that of the water vapor mixing ratio, depending on conditions."

11. What is the response time of your sensors, in particular humidity sensor of the radiosonde, and how was this compensated? Again, see publications of Bärfuss.

Response 11:

See our response to Comment 2 where we addressed this issue. The RH sensor associated with the dropsondes is sufficiently fast in the temperature range of the comparisons, making the effect of a time response correction minimal.

12. You state that the altitude plays a major role. How do you determine altitude? It says in l. 208 that it is geopotential height. So probably it was calculated based on pressure data. What is the accuracy of the pressure sensor, in particular for the radiosonde? What does this mean in terms of uncertainty for the altitude?

Response 12:

The geopotential altitude is integrated from the surface upward. To do so, ASPEN extrapolates the last measured pressure down to the surface using the last measured fall rate and the average time between the last complete data frame received and the time the sonde reaches the surface (and stops transmitting). The geopotential height is then integrated upward using this surface pressure. Therefore, the uncertainty of the pressure sensor directly affects the estimate of the surface pressure, but not the calculated geopotential height, assuming the calibration error is constant throughout the profile. The uncertainty of the geopotential height calculation, however, is determined by the surface extrapolation since the algorithm has no knowledge where within the last incomplete data frame the telemetry transmission ended. At a fall rate of 10 m s$^{-1}$ and a data rate of 2 Hz, we can conservatively assume an uncertainty of the geopotential height of less than 5 m. This uncertainty is an offset that affects the entire profile. We felt no changes were needed to the article file for this comment.

13. You state that RH is strongly dependent on temperature. So why don't you use a different parameter for humidity, like the water vapor mixing ratio, which is independent of temperature?

Response 13:

Thank you for your comment. We agree that the water vapor mixing ratio (WVMR), being independent of temperature, is a robust parameter for representing humidity. To address this, we included a direct comparison of WVMR between DLH and dropsonde data. The results in the new Figure 9f show a strong correlation (r = 0.84) and illustrate the consistency of WVMR measurements across instruments, regardless of temperature differences.

At the same time we feel that relative humidity (RH) remains an essential metric in our study due to its relevance to cloud microphysics, saturation processes, and phase transitions, all of which are inherently temperature-dependent. Figure 9d demonstrates that RH measurements also correlate strongly between the instruments (r = 0.91), validating its use in this context.

We formally address this comment with a revised form of Figure 9, which includes two new panels (e-f) to show the WVMR intercomparison to go along with the RH intercomparison in panels (c-d). Here is the associated text discussing the two new panels in the draft:

"Given that RH is temperature-dependent, we also examined water vapor mixing ratio (WVMR). With relaxed vertical criteria, the correlation between DLH and dropsonde WVMR values remained strong (r = 0.97; Fig. 9e), similar to the RH intercomparisons (r = 0.93). With stricter vertical criteria (Fig. 9f), the intercomparison was still strong (r = 0.84) and consistent with the RH intercomparisons (r = 0.91). This further confirms the robustness of our humidity measurements across different instruments, regardless of temperature differences."

[Figure]

Figure 9: (a) Temperature scatterplot between dropsonde and TAMMS pairs of single points following the method of Figure 1 (within 30 km, 15 min, and 25 m vertical spacing) excluding the last step to find the single points with the minimum altitude difference. (b) Same as (a) except showing only pairs following the full method of Figure 1 including the last step. (c-d) Same as (a-b), but for RH with markers colored by temperature difference: TAMMS – dropsonde. (e-f) Same as c-d, but for water vapor mixing ratio (WVMR).

14. The examples in Fig. 4 and 5 indicate highly dynamic conditions. Maybe they are not so suitable as an illustration?

Response 14:

These examples were intentionally chosen to illustrate the behavior of key parameters under dynamic atmospheric conditions. Dynamic cases highlight the variability and the ability of the instruments to capture rapid changes, which are critical for understanding the performance of the measurement systems in such scenarios.

While we acknowledge that these conditions may not represent calmer atmospheric situations, they are an essential part of the dataset and provide valuable insights into an intercomparison type of study for ACTIVATE. We believe that retaining these figures provides a realistic perspective on the challenges of conducting measurements in highly variable environments. As a result, we feel no changes are needed for the article in response to this comment.

Minor comments:

15. Differentiate between wind speed, wind direction and wind vector instead of simply using "wind" or even "winds".

Response 15:

Excellent point. We carefully reviewed the manuscript to replace generalized terms.

16. Do not use acronyms without explanation.

Response 16:

Thank you for your comment. We ensured that all acronyms are clearly defined upon their first use in the manuscript to improve clarity and accessibility for readers.

17. Put references in chronological order when citing them, e.g. l. 41, 42 Lewis and Schwartz, 2004, Nuijens and Stevens, 2012, Neukermans et al., 2018; throughout the manuscript.

Response 17:

Thank you for pointing this out. We revised the manuscript to ensure that all references are cited in chronological order throughout, including the example at lines 41–42.

**Reviewer 2:**

1: The paper gives a lengthy overview on previous results from airborne observation system intercomparisons. It is not made clear, for what purpose that is done. The cited results, such as wing-by-wing aircraft measurements, or comparisons between vertical profiles from dropsondes and wind lidar, are basically much different to those presented here. Almost all studies had a specific objective, such a comparison of a „new" technique to a proven one. All studies also aimed to keep spatial differences very small in the measurement setups. The present paper does not say, what their own study may provide as added value. Just claiming that „it has not been done before" does neither justify presenting that overview to be included.

Response 1:

Thank you for your feedback on the literature review and its role in our manuscript. We acknowledge the importance of ensuring that the overview of previous studies is directly tied to the motivation and contribution of our work. Although perhaps a matter of taste, we feel that providing context and literature review of past attempts to intercompare such systems is valuable for the big picture. We do not feel this hurts at all and can only help. More importantly though, what can probably help address most of this reviewer's concerns is a better explanation of the objective to convince them that we are not just doing this because "it has not been done before". We have taken steps to now say in different ways throughout the paper what our objective is to relieve such concerns. As summarized below, while this analysis is helpful for comparing independent measurements of similar variables (i.e., if they agree, that gives confidence in data quality), it importantly helps us gain confidence in data integration studies involving multiple spatially coordinated aircraft with one launching dropsondes; the latter has yet to be done in a report like ours. This is why we were motivated to do this paper including how there have been many questions surrounding the validity of this approach with the ACTIVATE dataset.

In Abstract:

"Airborne measurements of wind speed and direction, temperature (T), and relative humidity (RH) are critical due to their importance for atmospheric processes. Field campaigns with multiple coordinated aircraft present challenges when combining data from each platform due to atmospheric heterogeneity. To confront this issue, this work intercompares for the first time in situ measurements from the Turbulent Air Motion Measurement System (TAMMS) of horizontal winds and T, and a diode laser hygrometer (RH) deployed on a HU-25 Falcon flying mostly within the marine boundary layer over the northwest Atlantic to an independent set of measurements from dropsondes launched from a higher-flying King Air."

-and-

"Overall, these results provide confidence in both the performance of the measurement techniques compared and combining dropsonde data with in situ data from a separate coordinated aircraft for ACTIVATE, which has relevance to other campaigns with multiple coordinated aircraft conducting similar types of measurements."

In Introduction:

"Furthermore, a growing number of field campaigns like ACTIVATE, CAMP$^2$Ex, and ARCSIX involve multiple coordinated aircraft with one platform launching dropsondes that is spatiotemporally removed from other aircraft. The dropsonde data and other datasets from the same aircraft are used together with the other aircraft for various scientific applications, but the question remains as to how valid this exercise is due to heterogeneity in atmospheric conditions. Intercomparisons between atmospheric state variables between the different platforms can provide a view of how well this strategy can work. If such variable (e.g., winds, temperature, humidity) values from independent measurements agree well between the two aircraft in a complex environment like the northwest Atlantic (subject of this study), data from both aircraft can be used together in a meaningful way for certain scientific applications."

-and-

"If the results show good agreement, this provides confidence in both the performance of the independent measurement techniques and confidence in integrating dropsonde data with the Falcon for specific scientific applications especially if there are no Falcon data for atmospheric state variables on a given flight such as when icing impacts data quality."

In Methods section:

"Figure 1 summarizes the criteria applied to identify the best match between the Falcon aircraft measurements for a given dropsonde launched from the King Air altitude down to the ocean surface, **which ideally should be done for other campaigns involving multiple coordinated aircraft involving dropsonde launches.**"

In Conclusions:

"This effort is largely motivated by the importance of assessing airborne instrument performance and determining the validity to integrate dropsonde data with data collected from another coordinated aircraft, especially in light of a growing number of field campaigns with spatially coordinated aircraft."

-and-

"This work provides added confidence in data quality for the instruments compared and highlights the suitability of using dropsonde data launched from a higher flying aircraft to another one flying lower with a fair degree of spatial and temporal coordination for conditions such as the northwest Atlantic where there can be heterogeneity in atmospheric conditions."

2. The presentation of results in terms of statistical measures is a bit of a „technical report" style. The reader has little chance to follow the presentation and keeping an overview on all the given numbers of correlations, biases etc. This is particularly the case in the discussion of certain quantities like humidity or temperature. Both the quantative results as well as their interpretation must become more focussed.

Response 2:

Thank you for your insightful comment. We have streamlined the presentation of correlation values, biases, and other statistical metrics by summarizing key findings in tables rather than presenting long numerical sequences in the text. This makes it easier for the reader to not get lost in numbers and distractions in the main body of the manuscript and to make the choice for themselves if they want to refer to the tables to extract the quantitative results. Overall, the nature of this work is intercomparisons between atmospheric state variable values as measured by two different airborne platforms and thus naturally there needs to be statistical calculations conducted to do comparisons. We feel the text is fairly concise and focused but we still have tried to put a brighter spotlight on the focus with some minor revisions. We could move more materials to the Supplement but prefer not to at this stage.

3. The authors compare data from a sophisticated aircraft measurement system (TAMMS, laser diode hygrometer) and a widely used dropsonde system (NRD41). The aircraft instrumentation will have undergone the high quality standards of sensor selection, calibration and fusion (such as for wind) to obtain meteorological quantities. There are certain limitations, of course, such as when measurements are made during certain aircraft maneures. Excluding such cases as done, the general in-flight accuracy can be well estimated. Efficient ways to test for remaining problems, such as alignment errors after (re)installation of instruments, are normally detected by changing flight directions in homogeneous wind. Wing-by-wing flights have been also performed for many research aircraft, to detect - mostly small – systematic or random arrors. It is not pointed out, why „occasional" near-by-passes with dropsonde can be really used to detect errors in aircraft measurements.

Response 3:

We have revised the manuscript to state that our objective in this work is to help with questions that arise during a growing number of airborne field campaigns utilizing multiple aircraft as to whether it is fair to combine data from two or more aircraft claimed to be "coordinated" for scientific applications. We focus on comparing dropsondes launched from one aircraft to instruments on another aircraft to see if atmospheric state parameter values agree using specific spatiotemporal criteria explicitly noted in the text. The idea is that if the values agree well, this gives confidence to a growing number of data users that it is fair to combine data (and not just the state parameter data, but also other instrument datasets from the different aircraft) for scientific applications. But often the in situ measurements on one plane are not available (e.g., icing issues on the lower flying plane), which motivates the usage of the dropsonde data for key atmospheric state variables, which is why this work is important. The question has come up a lot even for ACTIVATE data users as to the validity of using dropsonde data along with Falcon data. This study aimed to comprehensively study this issue to help not just with ACTIVATE data usage but also a growing number of field campaigns using multiple coordinated aircraft. We are assessing the level of agreement in light of the heterogeneity that exists over the northwest Atlantic rather than framing this study as being one to characterize errors for any given instrument. We feel that with our text revisions to sharpen this focus for readers that this is sufficient to address this excellent comment.

Relevant revisions for his comment are shown in the response to Comment #1 above.

4. Similarly, a widely used and proven dropsonde like the NRD41 in general will not be subject to large or unknown systematic errors. There are exceptions, such as for humidity in clouds and in the upper troposphere, but lab calibrations, pre-flight checks, sonde intercomparisons ensure that problems are

mostly limited to indidual sondes, such as when a electronics component fails or a unfavourable GPS-constellation is given in a certain height range. A lot of information is also available from radiosonde intercomparisons, such as in the WMO UAII 2022 Campaign. Though not fully applicable to dropsondes, basic information on sensors accuracies is relevant for both for radiosondes and dropsondes, being partly quite similar in electronics, sensors and data handling. Again, the question arises what additional value is provided by measurements from an aircraft passing only occasionally a dropsonde.

Response 4:

The response to this comment is the same as the previous comment. We have sharpened the focus explicitly in words to alleviate such concerns. Collectively the authors of this study have been involved with dozens of field campaigns and we feel confident that this paper is valuable in the context of informing data users about the validity of using data between multiple coordinated aircraft together for scientific applications.

5. Atmospheric inhomogeneity is presumably a major reason for the differences between aircraft and dropsonde data. Ideal collocated measurements were not possible for reducing differences to measurements system differences. Correlation coefficients and scatter in biases, and other statistical metrics must be considered as functions of horizontal and vertical separation d. Only in the limiting case of d -> 0, knowledge about instrument differences can be obtained. The data sample ist big for large separations, however, but getting very small for small separations. This seems to exclude a robust extrapolation to r -> 0.

Response 5:

We agree that differences between the Falcon and dropsonde data can be influenced by how far apart (both spatially and temporally) the measurements are taken. To address this, we carefully looked at how both vertical and horizontal distance impact our comparisons. The purpose of this study thrives in a way that there are heterogeneities since we are evaluating how valid it is to intercompare data from multiple coordinated aircraft in projects like ACTIVATE. The results of this study are important to encourage data intercomparisons between dropsondes from the King Air and the Falcon for instance, whereas before such a study there was more uncertainty about the validity of doing this.

6. A remaining application option would be optimizing combinations of aircraft and dropsonde data to derive two-dimensional cross sections through the atmosphere. Excluding biases between dropsondes and aircraft data would be helpful for that purpose. But the authors do not address this issue.

Response 6: As noted in most of the responses above, we feel that the text revisions we made to explicitly mention our focus should alleviate concerns of this reviewer.

Minor issues

1.Due to the general criticism in the overall part, a detailed sentence-by-sentence

Response: No revisions needed for this comment since we were not sure what was meant here and if the sentence was inadvertently cut off somehow.

2.The reviewer did not review the English language, since there are enough native speakers in the not group.

Response: Thorough English editing has been done for the reasons stated.

3.The technical presentation (figures, tables, captions, citations, literature) seems to be consistent to AMT standards.

Response: No changes are needed for this comment.

---

## Referee Report (RR1)

**Intercomparison of wind speed, temperature, and humidity data between dropsondes and aircraft in situ measurements**

**Review Report**

The manuscript seems to be well structured and the study carried out is explained relatively well, most notably following the amendments implemented. Incidentally there are some improvements that can be implemented. Please note that although for each suggested change some examples from the manuscript have been reproduced here, not all instances of each suggested change are being highlighted in this report and the manuscript should be reviewed in its entirety for the amendments suggested below.

1. The term 'We' (first person plural) is being used throughout the manuscript. In scientific reporting the usual practise is to use the passive voice such as 'It should be emphasized...' instead of '...we emphasize...' (line 120) and 'Intercomparisons of RH were conducted...' instead of 'We conduct RH intercomparisons...' (line 154), and '...it was decided...' instead of '...we decided...' (line 160), etc. Please review for all the manuscript.

2. Line 163 – Should the term in brackets read 'Sect.' instead of 'Text'?

3. There are some grammatical errors which may result in the text being misinterpreted or misunderstood entirely. Although the following are some examples, there are other instances in the document where the sentence structure may be improved to enhance the readability and understanding of the document. The following are some examples:

   a. Line 178 – should the term used be 'compute' instead of 'computer'?

   b. Line 229 – 'We note that for some geophysical variables compared in this work, there were times when data were collected and thus there were not a full set of 555 pairs to compare.' This sentence maybe misinterpreted, rephrasing should be considered.

   c. Line 300 – '...which afford the only way with ACTIVATE in situ data to compare horizontal changes at a fixed vertical level...' this maybe misinterpreted, rephrasing should be considered.

   d. Line 378 – 'Figure 6 intercompares wind speeds from the TAMMS-dropsonde pairs two different ways.' This is not completely clear, rephrasing should be considered.

4. It has been also noticed that there is significant use of colloquial language. This is not the norm in scientific reporting and should be avoided:

a. Line 207 – I suggest that the phrase 'apples to apples' be replaced with a non-colloquial term/phrase such as 'appropriate'.

b. Line 398 – 'To dig deeper into wind speed intercomparisons, we stick...' the use of terms 'dig deeper' and 'stick' should be avoided. I suggest that alternative phrasing should be used.

c. Line 536 – 'Our results are comparable to what is reported in the literature (Table 1) although it is not an "apples to apples" comparison...' – rephrasing should be considered to avoid the use of colloquial language.

5. Line 461 – '**Intercomparison of T and RH**' – Parameter symbols should be avoided in titles and subtitles. This is also a question of writing style. One may use parameter symbols only in formulae, and always use the full parameter names in the text. For example in Line 474 'Relative humidity data values did not agree as well as T due...' the manuscript switches between full parameter names and parameter symbols in the same sentence. I suggest that consistency is maintained throughout the manuscript, as already suggested in this point. The entire manuscript should be reviewed to address this inconsistency.

6. Lines 536-553 should not form part of the conclusion. This part should be at the end of the 'Results and Discussion' section. The 'Conclusion' should only be reporting the main findings of the research being presented in the paper, and potentially any further research emanating from this study.

---

## Author Response (AR2)

We thank the reviewer for the thorough review of our manuscript. We have addressed each comment below and revised the manuscript accordingly. The paper now has improved clarity to allow for better reader comprehension.

**Response to Reviewer Comments**

**Reviewer 1:**

1: The term 'We' (first person plural) is being used throughout the manuscript. In scientific reporting the usual practise is to use the passive voice such as 'It should be emphasized…' instead of '…we emphasize…' (line 120) and 'Intercomparisons of RH were conducted…' instead of 'We conduct RH intercomparisons…' (line 154), and '…it was decided…' instead of '…we decided…' (line 160), etc. Please review for all the manuscript.

Response 1:

This adjustment has been applied consistently throughout the manuscript.

2. Line 163 – Should the term in brackets read 'Sect.' instead of 'Text'?

Response 2:  Change made.

3. There are some grammatical errors which may result in the text being misinterpreted or misunderstood entirely. Although the following are some examples, there are other instances in the document where the sentence structure may be improved to enhance the readability and understanding of the document. The following are some examples:

   a.  Line 178 – should the term used be 'compute' instead of 'computer'?
   b.  Line 229 – 'We note that for some geophysical variables compared in this work, there were times when data were collected and thus there were not a full set of 555 pairs to compare.' This sentence maybe misinterpreted, rephrasing should be considered.
   c.  Line 300 – '…which afford the only way with ACTIVATE in situ data to compare horizontal changes at a fixed vertical level...' this maybe misinterpreted, rephrasing should be considered.
   d.  Line 378 – 'Figure 6 intercompares wind speeds from the TAMMS-dropsonde pairs two different ways.' This is not completely clear, rephrasing should be considered.

Response 3:  We thank the reviewer for catching these problems, which we fixed. We have similarly and diligently gone through the rest of the manuscript to improve readability.

4. It has been also noticed that there is significant use of colloquial language. This is not the norm in scientific reporting and should be avoided:

   a.  Line 207 – I suggest that the phrase 'apples to apples' be replaced with a non-colloquial term/phrase such as 'appropriate'.

b. Line 398 – 'To dig deeper into wind speed intercomparisons, we stick...' the use of terms 'dig deeper' and 'stick' should be avoided. I suggest that alternative phrasing should be used.

c. Line 536 – 'Our results are comparable to what is reported in the literature (Table 1) although it is not an "apples to apples" comparison...' – rephrasing should be considered to avoid the use of colloquial language.

Response 4: All of these corrections have been made and we scrutinized the rest of the paper too in order to improve readability and adherence to more scientific terminology.

5. Line 461 – '**Intercomparison of T and RH**' – Parameter symbols should be avoided in titles and subtitles. This is also a question of writing style. One may use parameter symbols only in formulae, and always use the full parameter names in the text. For example in Line 474 'Relative humidity data values did not agree as well as T due...' the manuscript switches between full parameter names and parameter symbols in the same sentence. I suggest that consistency is maintained throughout the manuscript, as already suggested in this point. The entire manuscript should be reviewed to address this inconsistency.

Response 5: We fixed the Section title and all instances of symbols in the text. We sometimes use the variable symbols in tables (e.g., Table 2) where it helps to reduce letters in the column headers to keep the tables from getting too wide.

6. Lines 536-553 should not form part of the conclusion. This part should be at the end of the 'Results and Discussion' section. The 'Conclusion' should only be reporting the main findings of the research being presented in the paper, and potentially any further research emanating from this study.

Response 6: We moved the lines in question to a newly created subsection at end of Results and Discussion as the reviewer suggested (Section 3.4: Intercomparison to Literature).

---

## Author Response (AR3)

Editor Request for Correction:

As a minor typographical issue, note bold text on line 520:

"3.4 Intercomparison to Literature

This study's results are comparable to what is reported in the literature (Table 1) although it is not a direct", only the heading should be bold.

Response: We confirmed that the part after the heading was never in bold, and thus no change is needed.